# Unveiling Prior-Data Fitted Networks on Causal Effect Estimation: Pre-Training or Fine-Tuning?

**Haotian Wang** [1]  **Xinpeng Lv** [1]  **Hao Zou** [2]  **Yanghao Xiao** [3]  **Shanzhi Gu** [1]  **Yang Shi** [4]  **Yunxin Mao** [1]
**Yuanxing Zhang** [4]  **Mingyang Geng** [1]  **Shaowu Yang** [1]  **Haoxuan Li** [3][✉]  **Wenjing Yang** [1]  **Peng Cui** [2][✉]
**Zhouchen Lin** [3]

## Abstract

Amortized causal inference via Prior-data Fitted Networks (PFNs) has emerged as a promising paradigm, enabling zero-shot estimation of causal effects without the need for dataset-specific model tuning. However, the principled effectiveness of unified pre-training across general interventional regimes remains an underexplored question. In this paper, we investigate interventions on subsets of variables within Structural Causal Models (SCMs) and identify a fundamental theoretical limitation of current pre-training approaches. Theoretically, we prove that a single observational SCM induces an exponentially large space of interventional distributions, resulting in a phenomenon we term prior uncoverage. Consequently, this uncoverage yields a mismatch between the learned meta-prior and the true grounding prior, leading to unavoidable posterior inconsistency and estimation bias. To address this, we posit that fine-tuning is a fundamental necessity and propose a target-specific strategy named **P**oint-**W**ise **I**nterventional Fine-tuning (PWF), enabling the local generalization property. We further scale this approach via *Meta-Sampling Fine-tuning* (MSF) from a budgeted active learning perspective, thereby achieving uniform generalization on any interventional distribution.

## 1. Introduction

Causal inference is fundamental across numerous domains, including public policy, economics, and healthcare (Prosperi et al., 2020; Dahabreh & Bibbins-Domingo, 2024; Vanderschueren, 2024; Kou et al., 2025; 2024). As the golden standard, i.e., explicit intervention (e.g., Randomized Controlled Trials), are often prohibitively expensive or ethically infeasible, a central challenge in the field lies in estimating causal quantities from observational data, where observed confounding factors obscure true effects (Pearl, 2009; Rubin, 1974).

Under the assumption of ignorability (Imbens & Rubin, 2015), researchers have developed a wide array of specialized causal estimators over the past decades. By leveraging machine learning techniques, these methods have enabled remarkable progress in estimating non-linear causal effects (Athey et al., 2018; Künzel et al., 2019; Shalit et al., 2017; Shi et al., 2019; Yao et al., 2018). However, a fundamental limitation of these estimators is their reliance on isolated, single-dataset training, which precludes the amortization of inference capabilities across diverse domains (Robertson et al., 2025). Consequently, practitioners face the burden of bespoke model selection and tuning for each application, necessitating computationally expensive re-training for every new data generating process (DGP).

In the era of foundation models, amortized causal inference, grounded in Bayesian modeling of underlying DGPs, presents a promising avenue to address these limitations. By parameterizing a meta-prior over plausible causal mechanisms, this framework infers the posterior predictive distribution of causal quantities conditioned on observed evidence (Rubin, 1978; Hill, 2011). Recently, advances in Prior-data Fitted Networks (PFNs) have further mitigated the drawbacks of traditional Bayesian methods, such as the high computational cost of posterior sampling and restrictive prior specifications. Concretely, PFNs leverage transformer architectures pre-trained on large-scale synthetic DGPs to encode a rich prior, performing posterior predictive inference directly via in-context learning. For instance, CausalPFN (Balazadeh et al., 2025) and DoPFN (Robert-

---

[1]College of Computer, National University of Defense Technology, Changsha, China [2]Tsinghua University, Beijing, China [3]State Key Lab of General AI, School of Intelligence Science and Technology, Peking University, Beijing, China [4]School of Computer Science, Peking University, Beijing, China. Correspondence to: Haoxuan Li <hxli@stu.pku.edu.cn>, Peng Cui <cuip@tsinghua.edu.cn>.

*Proceedings of the 43rd International Conference on Machine Learning*, Seoul, South Korea. PMLR 306, 2026. Copyright 2026 by the author(s).

son et al., 2025) have demonstrated success in causal effect estimation with single treatment, enabling zero-shot posterior inference on arbitrary testing data without re-training. Moreover, a recent breakthrough further expands the capability and boundary of PFN-style models (i.e., LimiX model (Zhang et al., 2025)) by achieving unified structured-data modeling through query-based conditional prediction, substantially broadening the scope of in-context causal inference from conventional prediction to imputation and data generation within an unified model.

In this paper, we investigate a fundamental yet underexplored question within the rise of amortized causal models: *Can unified pre-training achieve unbiased, amortized causal effect estimation across general interventional regimes?* To address this, we examine interventions on subsets of variables within a Structural Causal Model (SCM) and the resulting causal quantities. A critical observation in this context is that a single observational SCM can induce an exponentially large number of interventional distributions, thereby distinguishing causality-oriented tasks (Robertson et al., 2025) from standard prediction tasks (Hollmann et al.; Ma et al., 2025a). Building on this insight, we prove that the meta-prior learned by a pre-trained PFN exhibits a risk of exponential prior uncoverage regarding these interventional distributions (see our Theorem 1). Consequently, this leads to a mismatch between the learned meta-prior and the grounding prior (see our Theorem 2), yielding unavoidable posterior inconsistency and estimation bias (see our Theorem 3). Thus, our theoretical framework characterizes the intrinsic risks of unified pre-training for causal inference, which escalate with the scale of the SCM.

Consequently, we posit that fine-tuning is a fundamental necessity to correct the inevitable prior mismatch in pre-trained causal models. To this end, two fine-tuning frameworks are designed to restore valid generalization. First, we introduce *point-wise interventional fine-tuning* (PWF) by adapting the model to specific target interventions, achieving the property of *local generalization* within a defined neighborhood of the fine-tuning distribution (see our Theorem 5). We further develop a *Meta-Sampling Fine-tuning* (MSF) strategy by activelying cover the interventional space, thereby ensuring robust, amortized inference towards arbitrary interventional distributions (see our Theorem 6).

Our main contributions are summarized as follows:

- **Unveiling the Risk of Prior Uncoverage in Amortized Causal Inference:** We identify a fundamental limitation where a single observational SCM induces an exponentially large interventional space, causing *exponential prior uncoverage* in pre-trained PFNs (Theorem 1). This mismatch between the learned meta-prior and the unavoidable posterior bias (Theorem 2 and 3).
- **Establishing Valid Generalization via Interventional**

**Fine-tuning:** To restore generalization, we propose two fine-tuning strategies: *Point-Wise Interventional Fine-tuning* (PWF), ensuring *local generalization* within the neighborhood of the target distribution (Theorem 5), and *Meta-Sampling Fine-tuning* (MSF), ensuring robust, amortized inference across arbitrary regimes (Theorem 6).
- **Experimental Validations:** Experimental results across synthetic and real-world data verify the effectiveness of both our theory framework and the proposed fine-tuning strategies.

## 2. Related Work

### 2.1. Non-unified Causal Estimators

To estimate causal quantities, typical statistical methods focus on balancing the confounder by using diverse strategies, including reweighting (Kuang et al., 2020), matching (Stuart, 2010), covariate balancing (Athey et al., 2018) or doubly robust estimations (Van der Laan et al., 2011). To overcome the model misspecification for the high-dimensional, non-linear data, a bunch of machine learning methods are further introduced, such as tree-based methods (Athey & Wager, 2019; Wager & Athey, 2018), regression-based learners (X-, S-, DR-, and RA-Learners) (Künzel et al., 2019), and neural network approaches (Yao et al., 2018; Shalit et al., 2017; Shi et al., 2019; Wang et al., 2022; 2023; 2022; 2025). However, in the era of foundation models, it is necessary and appealing to develop a unified, amortized causal models without frequently re-training the base models.

### 2.2. Unified Pre-training for Causal Inference

**Amortized Causal Inference.** Amortized methods aim to build foundation models for unified pre-training for downstream causal inference tasks, i.e., unleashing the potential of a single, pre-trained model to estimate causal quantities across diverse data generation process (DGPs). To be specific, such methods can be categorized into two classes: (1) *Discovery-based Amortized Approaches*, which identifies causal quantities by discovering the underlying causal graphs (Peters et al., 2014; Zheng et al., 2018; Ke et al., 2022; Khemakhem et al., 2021) and then calculating interventionals (Scetbon et al.; Lorch et al., 2022); (2) *End-to-end Amortized Approaches*, which directly pre-train a unified effect estimation model with unified prediction results on arbitrary DGPs (Nilforoshan et al., 2023; Bynum et al., 2025; Zhang et al., 2023).

**PFN-based Amortized Causal Effect Estimation.** In recent, the advances of Prior-Data Fitted Network (PFN) offers new opportunities for amortized causal effect estimations by following the outline of Bayesian causal effect estimation (Li et al., 2023; Oganisian & Roy, 2021). PFN-based amortized causal models pre-train Transformers on

fully synthetic data, deriving the posterior distribution of target causal quantities with learned prior distributions with in-context samples (Robertson et al., 2025; Balazadeh et al., 2025; Ma et al., 2025b). More specifically, CausalPFN (Balazadeh et al., 2025) and DoPFN (Robertson et al., 2025) have informed the potential of unified causal effect estimation in the context of single treatment with high-dimensional covariates, achieving dominant cross-data estimations in the inference stage. Moreover, (Ma et al., 2025b) considers the causal insufficient regime by introducing the instrumental variables during the pre-training phase. LimiX (Zhang et al., 2025) recently proposed a unified structured-data modeling framework based on query-conditioned prediction, covering unified causal inference within a single model. This line of work complements PFN-style amortized inference by showing that in-context adaptation can serve as a general interface for diverse structured-data tasks.

By contrast, our paper aims to inform that *unified pre-training for general causal effect estimations is challenging*, leading to risk of prior uncoverage and estimation bias. Instead, designing specific fine-tuning strategies can make up for the flaw of pre-training for PFN-based causal models.

## 3. Preliminaries: Prior-data Fitted Networks (PFN) for Causal Inference

### 3.1. Structural Causal Models and Interventions

**Structural Causal Models.** We consider a structural causal model (SCM) $M = \langle X, G, P_\epsilon \rangle$ over endogenous variables $X = \{X_1, \ldots, X_D\}$, where $G$ is a directed acyclic graph (DAG). Each variable follows a structural equation $X_i := f_i(X_{\mathrm{pa}(i)}, \epsilon_i)$, with parent set $\mathrm{pa}(i)$ and exogenous noise $\epsilon_i$. The joint noise distribution $P_\epsilon$, together with the structural assignments, induces the observational distribution over $X$ under the Markov property. Each variable is assumed to take values in a finite domain of size at most $d$.

**Interventions and Interventional Distributions.** Let $\mathcal{S}_{\mathrm{obs}}$ denote the set of observational distributions. Following hard interventions (Pearl, 2009), the set of interventional distributions is

$$\mathcal{S}_{\mathrm{int}} = \left\{ P^{\mathrm{do}(\tilde{X}=\tilde{x})} : \tilde{X} \subseteq X, \ \tilde{x} \in \mathcal{X}_{\tilde{X}} \right\}. \quad (1)$$

From the SCM perspective, interventions replace the corresponding structural equations by $X_i := \tilde{x}_i$ for $X_i \in \tilde{X}$, while leaving other equations unchanged. Interventional distributions are then obtained by propagating the exogenous noise through the resulting interventional SCM.

**Causal Queries.** Given a target variable set $W \subseteq X$, we consider causal queries defined as functionals of the induced

marginal distribution,

$$Q^W(P) = \int q(w) \, P_W(dw), \quad (2)$$

where $q : \mathcal{W} \to \mathbb{R}$ is bounded and measurable. This form covers common estimands such as average and conditional causal effects. Examples include *Average Treatment Effect (ATE):* $W = \{Y\}$, where $Y$ is the outcome variable, and the causal query is $Q^W(P_{\mathrm{int}}) = \mathbb{E}_{P_{\mathrm{int}}}[Y]$; *Conditional Average Treatment Effect (CATE):* $W = \{Y, Z\}$, where $Z$ is a conditioning covariate set and the causal query becomes $Q^W(P_{\mathrm{int}}) = \mathbb{E}_{P_{\mathrm{int}\mathrm{Do}(S=a)}}[Y \mid Z = z, S = a]$ with interventions as $S = \{a\}$ and the same holds for $T = \{a'\}$.

### 3.2. Causal Effect Estimation with PFNs

PFNs (Zhang et al., 2025; Robertson et al., 2025) amortize Bayesian posterior prediction of causal queries by learning from synthetic data generated under a prior over SCMs (see Fig. 1).

**Prior over SCMs.** Let $\mathcal{M}$ denote the space of SCMs and $\Lambda$ a prior over $\mathcal{M}$. Each $M \sim \Lambda$ induces a distribution $P_M \in \mathcal{P}$, where $\mathcal{P} = \mathcal{S}_{\mathrm{obs}} \cup \mathcal{S}_{\mathrm{int}}$. Training data are generated by first sampling $M$ and then drawing samples from $P_M$.

**Training Objective.** During pre-training, an SCM $M \sim \Lambda_0$ is sampled, followed by an in-context dataset $D_n \sim P_M$. The corresponding causal query $Q^W(P_M)$ is computed analytically or via simulation. The PFN $f_\theta$ is trained by minimizing

$$\mathcal{L}(\theta) = \mathbb{E}_{M, D_n}\left[ -\log f_\theta(Q^W(P_M) \mid D_n) \right]. \quad (3)$$

After pre-training, the PFN can be viewed as an empirical meta-prior $\hat{\Lambda} = \frac{1}{N} \sum_{i=1}^N \delta_{M^{(i)}}$ on a finite set of SCMs.

**Inference and Posterior Prediction.** At inference time, PFNs are provided with $n$ in-context samples $D_n$, which may include observational and/or interventional data. Given $\hat{\Lambda}$, the posterior predictive distribution of $Q^W$ is

$$\Pi^{Q^W}(B \mid D_n) = \int_{\mathcal{M}} \mathbb{I}(Q^W(P_M) \in B) \, \hat{\Lambda}(dM \mid D_n), \quad (4)$$

for any Borel set $B$, which can equivalently be expressed in the distribution space $\mathcal{P}$ induced by SCMs.

## 4. Theory: Risk of Pre-trained PFNs

### 4.1. Equivalent Class From SCM to Distributional Prior

We next inform that an equivalence relationship between $\mathcal{M}$ and $\mathcal{P}$, by defing a mapping $\Phi$ from $\mathcal{M}$ to $\mathcal{P}$: $\Phi : M \mapsto P_M$, where the distribution $P_M$ is induced by propagating $P_\epsilon$

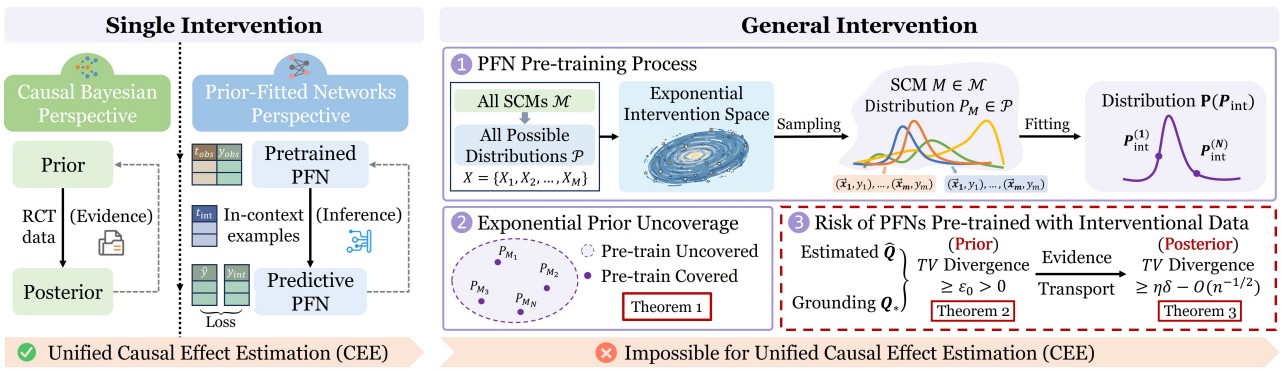

Figure 1. Left: Existing PFN-based causal foundation models on single intervention (treatment); Right: Our analysis on general intervention, which informs the uncoverage risk, prior mismatch and posterior bias.

through the SCM $M$ deterministically. Thus, $\Phi$ maps from $\mathcal{M}$ to $\mathcal{P}$, and we use $[M] := \{M' \in \mathcal{M} : \Phi(M') = P_M\}$ to denote the equivalence class of SCMs inducing the same distribution $P_M$[1].

**Lemma 1** (Push-forward Posterior from SCM to Distributions). *Let $\Lambda$ be any prior on $\mathcal{M}$, and let $\Pi := \Phi_\# \Lambda$ be its push-forward prior on $\mathcal{P}$, i.e. $\Pi(B) = \Lambda(\Phi^{-1}(B))$ for any $B \subseteq \mathcal{P}$. Define the corresponding posteriors as $\Lambda_n(A) := \frac{\int_A L(D_n|M)\,\Lambda(dM)}{\int_{\mathcal{M}} L(D_n|M)\,\Lambda(dM)}$ and $\Pi_n(B) := \frac{\int_B L(D_n|P)\,\Pi(dP)}{\int_{\mathcal{P}} L(D_n|P)\,\Pi(dP)}$. Then the posteriors satisfy $\Phi_\# \Lambda_n = \Pi_n$, i.e., for all measurable $B \subseteq \mathcal{P}$, $\Lambda(\Phi^{-1}(B) \mid D_n) = \Pi(B \mid D_n)$.*

Therefore, as Lemma 1 has proved the push-forward relationship between posterior update in the SCM space $\mathcal{M}$ and that in the distributional space $\mathcal{P}$, it is sufficient to derive the risk of pre-trained PFNs equipped with priors supported in $\mathcal{P}$ (Our corollary 8 informs this fact in the later subsection).

**Remark 1** (Analysis on Distributional Space). *Thus, we let the true meta-prior over these interventional distributions be $\Pi$, parameterized by a weight function $\pi(\tilde{X}, \tilde{x}) > 0$ satisfying $\sum_{\tilde{X} \subseteq X} \sum_{\tilde{x} \in \mathcal{X}_{\tilde{X}}} \pi(\tilde{X}, \tilde{x}) = 1$. Moreover, the learned, empirical meta-prior is updated as $\hat{\Pi} = \frac{1}{N} \sum_{i=1}^{N} \delta_{P^{(i)}}$, which covers only a finite subset of $\mathcal{P}$, denoted as $\overline{S}_{\text{int}} = \{P^{(1)}, \ldots, P^{(N)}\} \subset \mathcal{P}$. Moreover, we denote the uncovered intervention set be $S^p_{\text{zero}} = \mathcal{S}_{\text{int}} \setminus \overline{S}_{\text{int}}$.*

### 4.2. Uncovered Risk of Interventional Distributions

First, we inform the risk of a pre-trained PFN on uncovering interventional distributions when estimating causal effects during the inference stage. More specifically, the following theorem informs that such risk tends to become significant with increasing variable number of the underlying SCM.

**Theorem 1** (Exponential Prior Uncoverage under Finite Interventional Pretraining). *Let $\Pi$ be a meta-prior over $S_{\text{int}}$ such that $\pi(\tilde{X}, \tilde{x}) > 0$ for all $(\tilde{X}, \tilde{x})$. Suppose a PFN is pre-*

trained on a finite subset $\overline{S}_{\text{int}} = \{P^{(1)}, \ldots, P^{(N)}\} \subset S_{\text{int}}$, yielding the empirical prior $\hat{\Pi}$. Then the uncovered prior mass $\delta := \Pi(S^p_{\text{zero}}) = \Pi(S_{\text{int}} \setminus \overline{S}_{\text{int}})$ satisfies

$$\delta \geq 1 - \frac{N}{(1+d)^D}, \tag{5}$$

and in particular, for any polynomially bounded $N = \text{poly}(D, d)$, we have $\lim_{D \to \infty} \delta = 1$, i.e., the uncovered interventional prior mass converges to one exponentially fast in the number of variables $D$.

### 4.3. Bias Propagation: From Prior to Posterior

Subsequently, we then first quantify the divergence between the prior $\hat{\Pi}$ estimated by pre-trained PFNs and the grounding prior $\Pi$ in below, as shown in Fig. 1.

**Theorem 2** (Prior Divergence Lower Bound). *Let the total mass of $S^p_{\text{zero}}$ under the grounding prior $\Pi$ as $\delta = \sum_{P^{\text{do}(\tilde{X}=\tilde{x})} \in S^p_{\text{zero}}} \pi(\tilde{X}, \tilde{x})$. Moreover, suppose further that any uncovered interventional distribution $P^{\text{do}(\tilde{X}=\tilde{x})} \in S^p_{\text{zero}}$ has total-variation distance to the nearest covered one satisfying*

$$\inf_{Q \in \overline{S}_{\text{int}}} \text{TV}\left(P^{\text{do}(\tilde{X}=\tilde{x})}, Q\right) \geq \varepsilon_0 > 0. \tag{6}$$

*Then, the following universal lower bounds hold:*

$$\text{TV}(\Pi, \hat{\Pi}) \geq \delta, \qquad W_{\text{TV}}(\Pi, \hat{\Pi}) \geq \varepsilon_0 \, \delta. \tag{7}$$

Notably, in our Theorem 2, $\delta$ quantifies the total prior mass of the uncovered intervention space under $\Pi$, and $\varepsilon_0$ captures the minimal divergence gap between uncovered and covered distributions. Together, the above factors yield a distribution-free lower bound on the discrepancy between the pretrained empirical meta-prior $\hat{\Pi}$ and the true causal prior $\Pi$. In concrete, we analyze three typical SCMs to offer a deeper insight into Theorem 2:

---

[1]Each SCM $M$ only induces one distribution.

**Example 1** (Concrete Prior: Example). *The following example of **Uniform prior** $\Pi$ interprets the prior mismatch (see detailed examples in Appendix A.4). If $\pi(\tilde{X}, \tilde{x}) = \frac{1}{(1+d)^D}$, then the uncovered mass equals $\delta = 1 - \frac{N}{(1+d)^D}$, and thus*

$$\text{TV}(\Pi, \hat{\Pi}) \geq 1 - \frac{N}{(1+d)^D}, \quad (8)$$

Subsequently, we further quantify the updated posterior by PFNs and the grounding posterior, which further informs the gap between the estimated interventional query $\hat{Q}$ and the grounding $Q_*$:

**Theorem 3** (Posterior inconsistency under prior divergence). *For any functional of the interventional distribution*

$$Q(P_{\text{int}}) = \int q(x) \, P_{\text{int}}(dx), \quad (9)$$

*let $\Pi_N(Q)$ denote the posterior of $Q$ under $\hat{\Pi}$ after observing $N$ samples from the causal system. Assuming that:*
*(1) The causal queries $Q, Q^\star$ and $P_{\text{int}}$ satisfies the standard Bernstein–von Mises regularity conditions;*
*(2) The queried interventional statistics distinguish distributions in $S^p_{\text{zero}}$ from its complement, i.e., there exists a constant $\eta > 0$ such that*

$$\inf_{P \in S^p_{\text{zero}}} \inf_{P' \in \overline{S}_{\text{int}}} \left| Q(P) - Q(P') \right| \geq \eta \|P - P'\|_{TV}. \quad (10)$$

*(3) The function $q : \mathcal{X} \to \mathbb{R}$ is measurable and bounded.*

*Then the posterior contraction rate satisfies*

$$\|\Pi_N(Q) - \mathcal{N}(Q_*, I^{-1}(Q_*)/N)\|_{\text{TV}} \geq \delta - O(n^{-1/2}), \quad (11)$$

*and the interventional query induced by $\hat{\Pi}$ admits the lower bound:*

$$\mathbb{E}_{P_{\text{int}} \sim \hat{\Pi}} \left[ |Q(P_{\text{int}}) - Q_*| \right] \geq \eta\delta - O(n^{-1/2}), \quad (12)$$

*which implies that the posterior cannot asymptotically converge to the nominal Gaussian limit unless $N \gg \delta^{-2}$ or $\text{TV}(\Pi, \hat{\Pi}) \to 0$.*

Theorem 3 (bias in posterior) together with Theorem 2 (bias in prior) informs that the risk of interventionally pre-trained PFN models comes from the existence of **interventional distributions $P_{\text{int}}$ uncovered by the pre-training phase, with enough dissimilarity of $P_{\text{int}}$ from covered distributions.**

Consequently, the push-forward relationship in Lemma 1 between $\mathcal{M}$ and $\mathcal{P}$ informs that our conclusion in Theorem 3 fits back to the **practical pre-training protocols of PFNs:**

**Corollary 4** (Posterior TV gap lifts from $\mathcal{P}$ to $\mathcal{M}$). *Let $\Lambda, \hat{\Lambda}$ be two priors on $\mathcal{M}$, and $\Pi = \Phi_\# \Lambda$, $\hat{\Pi} = \Phi_\# \hat{\Lambda}$ their induced priors on $\mathcal{P}$. Let $\Lambda_n, \hat{\Lambda}_n$ and $\Pi_n, \hat{\Pi}_n$ be the respective posteriors given $D_n$. Then for some sequence $n$,*

$$\|\Lambda_n - \hat{\Lambda}_n\|_{\text{TV}} \geq \|\Pi_n - \hat{\Pi}_n\|_{\text{TV}} \geq \delta - O(n^{-1/2}). \quad (13)$$

# 5. Local Generalization of Interventionally Fine-tuned PFN Models

Let $S = \{X_1, \ldots, X_k\} \subseteq X$ and $T = \{X_1, \ldots, X_m\} \subseteq X$ denote two (possibly distinct) intervention sets, with corresponding interventional distributions $P^S_{\text{int}} := P(X \mid \text{do}(S = s))$ and $P^T_{\text{int}} := P(X \mid \text{do}(T = t))$.

## 5.1. Local Generalization under Point-wise Interventional fine-tuning

As fine-tuning the PFN model using point-wise interventional distribution follows the standard protocol of model tuning, we focus on analyzing an important property named "**local generalization**". Intuitively, local generalization refers to the generalization capability of tuned PFNs when the causal query comes from the neighboring set of $P^0_{\text{int}}$, i.e., the interventional distribution used for fine-tuning.

**Theorem 5** (Local Generalization of PWF). *Let $P^0_{\text{int}}$ denote the point-wise interventional distribution for fine-tuning, and let $\mathcal{B}_\varepsilon(P^0_{\text{int}}) := \{P : \text{TV}(P_W, P^{0,W}_{\text{int}}) \leq \varepsilon\}$ denote the TV ball of radius $\varepsilon$ around $P^0_{\text{int}}$ in the marginal $W$-space. As the PFN (parametrized by $\theta^t$ in the round $t$ of fine-tuning) is micro-tuned at step $t$ via empirical samples by sampling $= \{X^{(i)}\}^{n_f}_{i=1}$ from $P^0_{\text{int}}$ and optimizing $q(W \mid \theta^t)$. Then, with probability at least $1 - \delta$ over the sampling of $X^{(t)}$, and assuming that $Q$ is $M_q$-Lipschitz, the local gereralization capability of tuned PFN holds for any interventional distribution $P$ with distance from $P^0_{\text{int}}$ is exhibited in below:*

$$\sup_{P \in \mathcal{B}_\varepsilon(P^0_{\text{int}})} \left| \hat{Q}_t - Q^W(P) \right| \leq$$

$$\underbrace{\Delta^t_{\text{opt}}}_{\text{optimization bias}} + \underbrace{M_q \sqrt{\frac{2 \log(2/\delta)}{n_f}}}_{\text{sampling error}} + \underbrace{2M_q \varepsilon}_{\text{TV-ball drift}}, \quad (14)$$

*where $\Delta^t_{\text{opt}} := |Q^W(P^0_{\text{int}}) - Q^W(q(W \mid \theta^t))|$ is the optimization bias[2].*

**Remark 2** (Lift Back to SCM Prior). *We inform that it is sufficient to analyze in the distributional-prior space $\mathcal{P}$ rather than in the SCM-prior space $\mathcal{M}$. More specifically, as Lemma 1 already informs the equivalence-class relationship from $\mathcal{M}$ to $\mathcal{P}$, one can easily extend the supremum over $P \in B$ to the supremum over $M \in \Phi^{-1}(B)$, i.e., the supremum in the SCM space, with the boundness of the supremum still holds.*

## 5.2. Local Generalization under Meta-Sampling Fine-tuning

As shown in Fig. 2, we then develop the second fine-tuning paradigm, namely the **Meta-Sampling Fine-tuning (MSF)**

---

[2]See more concrete examples in Appendix B.2

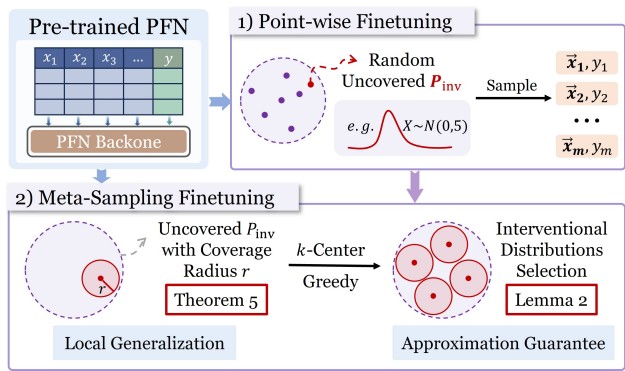

*Figure 2.* Illustration of our funetuning strategies.

approach, aiming to improve generalization capability over the space of interventional distributions.

**Active Budgeted Interventional Selection.** Let $\mathcal{S}_{\text{int}}$ denote a (possibly large) candidate set of interventional distributions. Given a sampling budget $K \ll |\mathcal{S}_{\text{int}}|$, MSF aims to *actively select* a subset $\mathcal{S} = \{P_{\text{int}}^{(1)}, \ldots, P_{\text{int}}^{(K)}\} \subseteq \mathcal{S}_{\text{int}}$, and fine-tune the PFN by drawing samples from these $K$ distributions in a mixed fashion. Throughout, all interventional distributions are compared through their marginals on the shared target variable set $W$. This naturally casts MSF as a budgeted active learning problem (Li et al., 2022; Vazirani, 2001; Hacohen et al., 2022; Tsang et al., 2005) over the space of interventional distributions.

**Coverage Radius from Theorem 5.** A natural strategy is therefore to *select them so that their associated TV neighborhoods jointly cover the entire candidate family $\mathcal{S}_{\text{int}}$ as well as possible in the $W$-marginal space* (Qin et al., 2021; Sundin et al., 2019; Li et al., 2022), and the quality of MSF is governed by the coverage radius $\varepsilon(\mathcal{S})$ defined in (14). Thus, the generalization error of a single interventional distribution $P_{\text{int}}^0$ serves as a radius, yielding the *distributional core-set selection* problem (Qin et al., 2021; Vazirani, 2001) on $W$-marginals:

$$\mathcal{S}^\star \in \sup_{P \in \mathcal{S}_{\text{int}}} \min_{P' \in \mathcal{S}} \text{TV}(P_W, P'_W) \text{ for } \mathcal{S} \subseteq \mathcal{S}_{\text{int}}, |\mathcal{S}| \le K.$$
(15)

**Distributional predictions of $W$-marginals from PFNs.** We leverage the predictive structure of the PFN to construct *pseudo predictions* of $W$-marginals. For each candidate interventional distribution $P \in \mathcal{S}_{\text{int}}$, the pre-trained PFN induces a predictive distribution $q_\theta(W \mid \mathcal{D}_P)$ as a model-based approximation to the true marginal $P_W$, where $\mathcal{D}_P$ denotes a small context dataset associated with $P$.

**Approximation Guarantee.** Although the exact solution to (15) is NP-hard, a simple greedy algorithm that sequentially adds the farthest point from the current set yields a constant-factor approximation.

**Lemma 2** (*k*-Center Approximation Guarantee). *Let $\mathcal{S}_{\text{greedy}}$ be the set of $K$ interventional distributions selected by the greedy k-center algorithm under the distance $d(P, P') = \text{TV}(P_W, P'_W)$. Then the induced coverage radius satisfies $\varepsilon(\mathcal{S}_{\text{greedy}}) \le 2\varepsilon(\mathcal{S}^\star)$, where $\mathcal{S}^\star$ denotes an optimal solution to* (15).

**Uniform Generalization Capability of MSF.** We now connect the core-set selection principle to the generalization behavior of MSF:

**Theorem 6** (Uniform Generalization under MSF). *Let $\mathcal{S}_{\text{greedy}} \subseteq \mathcal{S}_{\text{int}}$ be a set of $K$ interventional distributions selected by the greedy k-center algorithm under the distance $d(P, P') = \text{TV}(P_W, P'_W)$, and let $\varepsilon(\mathcal{S}_{\text{greedy}})$ denote its induced coverage radius. Suppose that the PFN is fine-tuned using $n_f$ samples drawn from the mixture distribution supported on $\mathcal{S}_{\text{greedy}}$. Then, with probability at least $1 - \delta$, for any interventional distribution $P \in \mathcal{S}_{\text{int}}$, the following bound holds:*

$$\left|\widehat{Q}_t - Q^W(P)\right| \le \Delta_{\text{opt}}^t + M_q\sqrt{\frac{2\log(2/\delta)}{n_f}} + 4M_q\,\varepsilon(\mathcal{S}^\star),$$
(16)

*where $\mathcal{S}^\star$ denotes an optimal solution to the k-center objective in* (15).

**Consequence of Approximate Coverage.** Combining Theorem 6 of our MSF strategy further guarantees the generalization over the whole interventional distribution space $P \in \mathcal{S}_{\text{int}}$. In other words, MSF enjoys a principled and explicit generalization guarantee over the entire candidate family $\mathcal{S}_{\text{int}}$.

## 6. Experiments

### 6.1. Experimental Setup

**Datasets.** For synthetic data, we consider three types of SCM models, including the linear SCM, non-linear additive SCM, and interaction SCM (strong interactions among $X$):

- **Linear SCM:** The outcome is a linear combination of features $Y = \mathbf{X}^\top \beta + \varepsilon_Y, \quad \varepsilon_Y \sim \mathcal{N}(0, 0.1^2)$.
- **Non-linear Additive SCM:** To test the model's ability to handle non-linearity without complex feature dependencies, we define the outcome as:

$$Y = X_0 X_1 + \tanh\Big(\sum_{k=2}^{i} X_k\Big) + \sin\Big(\sum_{k=i+2}^{i+1+j} X_k\Big) + \varepsilon_Y,$$

where $\varepsilon_Y \sim \mathcal{N}(0, 0.1^2)$ and $(i, j)$ are the corresponding indices for the chosen dimension of variable $m$ (e.g., $i = \lfloor m/2 \rfloor$, $j = m - i - 2$). This model incorporates heterogeneous non-linear effects over multiple feature subsets while avoiding dense cross-dimensional interactions.

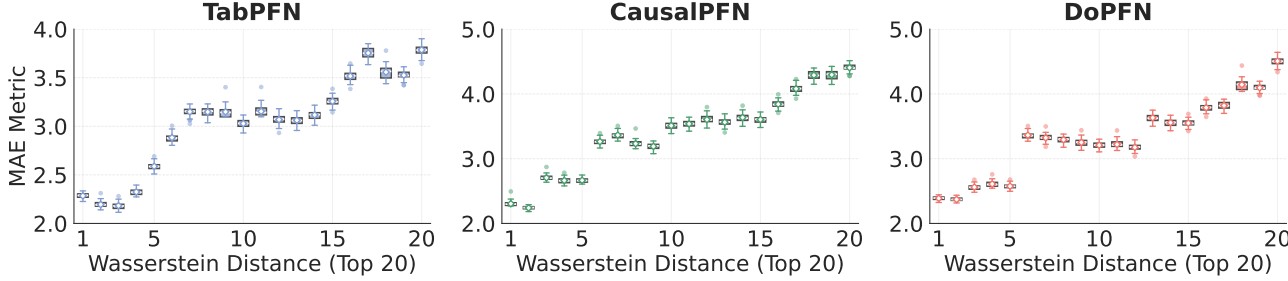

*Figure 3.* Local generalization of PWF across Nonlinear, Non-linear, and Interaction simulations, together with the Lalonde$_{\text{PSID}}$ benchmark. $D_1$ is the uncovered distribution used for fine-tuning. $D_2$ and $D_3$ represent distributions with low and high TV-distance from $D_1$, respectively.

*Figure 4.* Local Generalization of our PWF strategy on the Lalonde Dataset for each baseline, where x-axis refers to the testing interventional distributions with Top-K small divergence from the fine-tuned distribution.

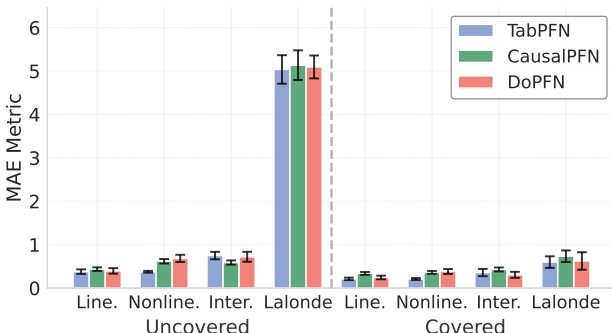

*Figure 5.* Counterfactual predictions of TabPFN, CausalPFN, and DoPFN across Uncovered and Covered interventional distributions for Linear, Nonlinear, Interaction simulations, and Lalonde.

- **Interaction SCM:** To simulate complex, high-dimensional dependencies across interventions, we consider the SCM with strong pairwise interactions:

$$Y = \alpha^\top X + \sum_{i<j} \gamma_{ij} X_i X_j + \varepsilon_Y, \ \varepsilon_Y \sim \mathcal{N}(0, 0.1^2).$$

For real-world datasets, we leveraged the RealCause framework (Neal et al., 2020) based on the Lalonde study to construct a semi-synthetic data generation pipeline, named Lalonde$_{\text{PSID}}$ (see details in Appendix C.4).

**Evaluation.** During the inference stage, we randomly select the corresponding, uncovered interventional distribution to assess our PWF strategy, together with extra distributions near/far from the selected distribution to evaluate the local

generalization property. To further assess our MSF strategy, we uniformly feed each uncovered distribution in $\mathcal{S}_{\text{int}}$ and report the maximum with average error. Concretely, we adopt the MSE and MAE metrics to evaluate the counterfactual prediction results under each intervention arms (uncovered by pre-training).

**Baselines.** For existing PFN-based causal models, we choose the original TabPFN model (Hollmann et al.), the in-context pre-trained DoPFN model (Robertson et al., 2025), and the equivalent posterior-based CausalPFN model (Balazadeh et al., 2025) (see detailed implementation in Appendix C). Regarding our fine-tuning approaches, upon three baselines, we further select specific point-wise interventionals by our PWF strategy, and using the $K$-greedy strategy to select the budget by our MSF strategy, where error bars represent $\pm$ one standard deviation.

Throughout our experiments, we aim to explore three questions: (1) How existing pre-trained PFN models perform on uncovered interventional distributions? (2) Will our PWF strategy achieve local generalization property? (3) Will our MSF strategy achieve near-uniform generalization property over all interventional distributions?

## 6.2. RQ1: Risk of Pre-trained PFN Models on Uncovered Interventionals

More specifically, we first examine the performance of pre-trained PFN models (with a subset of $\mathcal{S}_{\text{int}}$) on covered and uncovered interventional distributions. As illustrated in Fig-

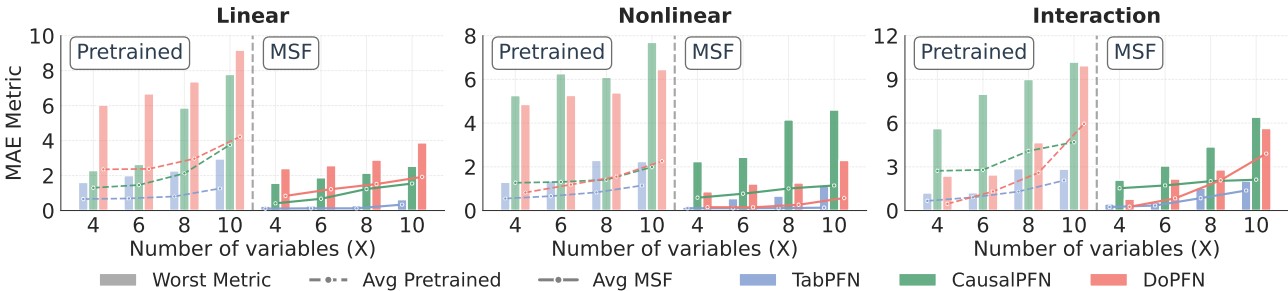

*Figure 6.* Uniform generalization assessment of the Meta-Sampling Fine-tuning (MSF) strategy across different variable scales. The figure compares counterfactual prediction results (MAE) against the MSF strategy over the entire set of uncovered interventionals.

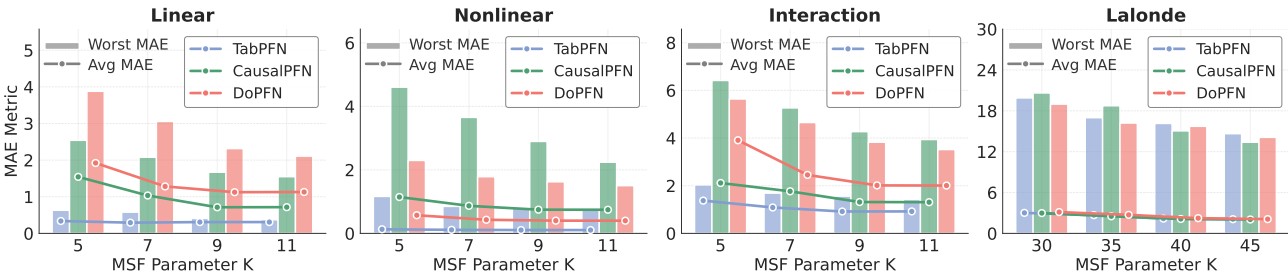

*Figure 7.* Performance evaluation of the the MSF strategy w.r.t the sampling budget size $K$ (at $|X| = 10$). The plots track the average and worst-case MAE for TabPFN, CausalPFN, and DoPFN across different structural causal mechanisms.

ure 5, in the *Covered* setting, all models maintain relatively low error rates, with TabPFN generally achieving the lowest MSE, particularly in Nonlinear scenarios (MSE $\approx 0.1$). However, in the *Uncovered* setting, the predictive risk increases substantially for all models (e.g., with significant enlarged error $> 5$ on the Lalonda dataset). Conversely, CausalPFN and DoPFN exhibit high variance and higher mean error in the Uncovered Nonlinear setting compared to their performance in covered distributions. These results suggest that while pre-trained PFNs excel at in-distribution interventional reasoning, they struggle to generalize to uncovered distributions with degraded causal estimations, verifying our theory in Section 4.

### 6.3. RQ2: Local Generalization Property of PWF

In this section, we evaluate the local generalization capabilities of the Point-wise fine-tuning (PWF) strategy. As illustrated in Figure 3, several key observations emerge: *(1) Significant Error Reduction on $D_1$:* Across all three structural settings (Linear, Nonlinear, and Interaction), applying PWF on the uncovered distribution $D_1$ leads to a dramatic decrease in MSE compared to the zero-shot "Pretrained" performance. *(2) Effective Local Generalization to $D_2$:* When evaluated on $D_2$ proximal to the fine-tuning distribution $D_1$, the MSE remains consistently low across all models, incidating indicates that the PWF strategy successfully captures the local interventional neighborhood. *(3) Limits of Local Generation on $D_3$:* The performance on $D_3$ (i.e., in-

terventionals far from $D_1$) is generally worse than on $D_2$, suggesting that using PWF can only cover limited range of generalized counterfactual prediction on interventionals.

Moreover, Figure 4 further reports the MSE as a function of the Wasserstein divergence between the fine-tuning and test interventional distributions. As the Wasserstein divergence increases, the error grows smoothly, reflecting the inherently local nature of point-wise fine-tuning.

### 6.4. RQ3: Uniform Generalization Property of MSF

To further evaluate the proposed MSF approach, we examine its capability to control the uniform generalization over the entire candidate set of interventional distributions $\mathcal{S}_{\text{int}}$. As shown in Figure 6, in the **Pretrained** (zero-shot) setting, all models exhibit a sharp escalation in predictive risk as the SCM excels, particularly in the Interaction setting where the worst-case error for CausalPFN exceeds 8.0 at $|X| = 10$. In contrast, our MSF yields to a steady error reduction in both **average MSE** and **worst-case MSE** for all considered backbones, informing the robust, amortized inference towards arbitrary interventional distributions.

Meanwhile, as illustrated in Figure 7, we analyze the impact of the **sampling budget** $K$ by observing a consistent trend where increasing the budget size $K$ from 5 to 11 for synthetic data and from 30 to 35 to the real-world Lalonde data. Notably, the TabPFN backbone maintains the most stable error profile, while CausalPFN and DoPFN exhibit

*Table 1.* Trends of the parameter $k_{\text{interv}}$ under different SCMs.

| Method | SCM | $k_{\text{interv}} = 2$ | $k_{\text{interv}} = 4$ | $k_{\text{interv}} = 5$ | $k_{\text{interv}} = 6$ | $k_{\text{interv}} = 7$ |
|---|---|---|---|---|---|---|
| TabPFN | Linear | 0.3087 | 0.3339 | 0.3518 | 0.3696 | 0.3824 |
| | Nonlinear | 0.1050 | 0.1367 | 0.1543 | 0.1728 | 0.1856 |
| | Interaction | 0.9183 | 1.0825 | 1.1684 | 1.2546 | 1.3182 |
| CausalPFN | Linear | 0.7134 | 0.7621 | 0.7968 | 0.8315 | 0.8582 |
| | Nonlinear | 0.7472 | 0.8163 | 0.8547 | 0.8926 | 0.9214 |
| | Interaction | 1.3157 | 1.4526 | 1.5228 | 1.5934 | 1.6487 |
| DoPFN | Linear | 1.1244 | 1.1986 | 1.2369 | 1.2754 | 1.3068 |
| | Nonlinear | 0.4012 | 0.4718 | 0.5096 | 0.5487 | 0.5783 |
| | Interaction | 2.0135 | 2.2714 | 2.3862 | 2.5028 | 2.5897 |

*Table 2.* Connection Between Theoretical Quantities and Empirical PWF Error

| TV Radius $\epsilon$ | Lipschitz $M_q$ | Coverage $r_{\text{cov}}$ | Linear MAE | Nonlinear MAE | Interaction MAE |
|---|---|---|---|---|---|
| 0.2 | 0.05 | 0.2 | 0.134 | 0.098 | 0.425 |
| 0.2 | 0.10 | 0.5 | 0.138 | 0.102 | 0.445 |
| 0.2 | 0.15 | 0.8 | 0.142 | 0.106 | 0.470 |
| 0.5 | 0.05 | 0.2 | 0.140 | 0.103 | 0.455 |
| 0.5 | 0.10 | 0.5 | 0.145 | 0.108 | 0.480 |
| 0.5 | 0.15 | 0.8 | 0.150 | 0.112 | 0.505 |
| 1.0 | 0.05 | 0.2 | 0.146 | 0.108 | 0.485 |
| 1.0 | 0.10 | 0.5 | 0.152 | 0.113 | 0.510 |
| 1.0 | 0.15 | 0.8 | 0.158 | 0.118 | 0.535 |

*Table 3.* Trends of the parameter $k_{\text{interv}}$ under different SCMs.

| Method | SCM | $k_{\text{interv}} = 2$ | $k_{\text{interv}} = 4$ | $k_{\text{interv}} = 5$ | $k_{\text{interv}} = 6$ | $k_{\text{interv}} = 7$ |
|---|---|---|---|---|---|---|
| TabPFN | Linear | 0.3087 | 0.3339 | 0.3518 | 0.3696 | 0.3824 |
| | Nonlinear | 0.1050 | 0.1367 | 0.1543 | 0.1728 | 0.1856 |
| | Interaction | 0.9183 | 1.0825 | 1.1684 | 1.2546 | 1.3182 |
| CausalPFN | Linear | 0.7134 | 0.7621 | 0.7968 | 0.8315 | 0.8582 |
| | Nonlinear | 0.7472 | 0.8163 | 0.8547 | 0.8926 | 0.9214 |
| | Interaction | 1.3157 | 1.4526 | 1.5228 | 1.5934 | 1.6487 |
| DoPFN | Linear | 1.1244 | 1.1986 | 1.2369 | 1.2754 | 1.3068 |
| | Nonlinear | 0.4012 | 0.4718 | 0.5096 | 0.5487 | 0.5783 |
| | Interaction | 2.0135 | 2.2714 | 2.3862 | 2.5028 | 2.5897 |

significant gains in predictive accuracy as the budget $K$ increases, entailing the necessity of choosing inappropriate $K$ with trade-off between budget cost and performance.

### 6.5. Impact of Varying Budget Size

We add additional experiments by varying the number of intervened variables to evaluate the scaling behavior under multi-variable interventions. Specifically, we vary the number of intervened variables $k_{\text{interv}} \in \{2, 4, 5, 6, 7\}$ while fixing the total number of variables as $K = 8$. The results are reported in Table 1. The results show that performance degrades gradually as the number of intervened variables increases, which is expected since the intervention space becomes more complex. Nevertheless, our MSF-based method remains relatively robust under multi-variable interventions, maintaining stable performance with limited accuracy degradation across different SCM settings.

### 6.6. Validations of Key Quantities in Our Theorems

We further report the empirical ranges of the key quantities appearing in our theoretical bounds and connect them with the observed prediction error. In the synthetic experiments, the TV-ball radius is controlled by the prescribed distribution-shift level, with $\epsilon \in [0.05, 1.0]$, evaluated at $\{0.05, 0.2, 0.5, 0.8, 1.0\}$. The Lipschitz factor lies in $M_q \in [0.05, 0.15]$ across different synthetic SCMs, where $M_q = 0.05$ for the linear SCM, $M_q = 0.07$ for the nonlinear additive SCM, and $M_q = 0.11$ for the interaction SCM. For MSF, the effective coverage radius of the selected $k$-center set lies in the range $r_{\text{cov}} \in [0.2, 0.8]$ under our experimental settings. As shown in Table 2, the MAE of PWF increases as the TV-ball radius, Lipschitz factor, and coverage radius become larger. This empirical trend is consistent with our theoretical result, suggesting that the bound captures meaningful performance degradation under larger distributional shifts and weaker coverage.

### 6.7. Interventions on Multi-variables in Simulations

We add additional experiments by varying the number of intervened variables to evaluate the scaling behavior under multi-variable interventions. Specifically, we vary the number of intervened variables as $k_{\text{interv}} \in \{2, 4, 5, 6, 7\}$ while

fixing the total number of variables as $K = 8$. The results are reported in Table 3. Our MSF-based method remains robust under multi-variable interventions, maintaining stable performance with limited accuracy degradation across different SCM settings.

## 7. Conclusion

We show that unified pre-training in Prior-data Fitted Networks (PFNs) is fundamentally insufficient for general causal effect estimation, as exponentially large interventional space cannot be covered by finite pre-training. This prior uncoverage leads to unavoidable posterior inconsistency and systematic estimation bias, distinguishing causal inference from standard prediction tasks. To resolve this, we demonstrate that interventional fine-tuning is necessary and propose Point-Wise Interventional Fine-tuning (PWF) for local generalization and Meta-Sampling Fine-tuning (MSF) for uniform coverage of the interventional space. Extensive experiments on synthetic and real-world benchmarks validate our theory and show that fine-tuning restores robust amortized causal inference.

## Acknowledgement

This work was jointly supported by the National Natural Science Foundation of China (Nos. 62276004, 62325604, 62525213), Tsinghua University-Siemens Joint Research Center (JCIIOT), the Beijing Natural Science Foundation (No. L257007), the Beijing Major Science and Technology Project (No. Z251100008425006), the Provincial Natural Science Foundation of Hunan Province (No. 2025JJ10008), the NUDT Youth Independent Innovation Science Fund (No. ZK25-20), and the State Key Laboratory of General Artificial Intelligence.

## Impact Statement

This paper aims to advance machine learning for amortized causal effect estimation, where pre-trained foundation models are expected to infer causal quantities across diverse interventional regimes without repeated model training. Such problems are central to socially relevant domains, including healthcare, public policy, economics, and resource allocation, where reliable estimation of intervention effects can support evidence-based decision-making. By identifying the prior uncoverage risk of PFN-based causal models and proposing interventional fine-tuning strategies to improve local and uniform generalization, our work may help reduce estimation bias caused by insufficient coverage of target interventional distributions.

At the same time, causal decision systems may have important societal implications. If the underlying causal assumptions are misspecified, if the available interventional or observational data are biased, or if the target population differs substantially from the pre-training and fine-tuning distributions, the resulting causal estimates may still be unreliable and may lead to unfair or harmful decisions across groups.

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

# Appendix

# A. Proof of Theories: Risk of Unified Pre-training

## A.1. Theory of Push-forward Relationship between SCM Prior and Distributional Prior

**Lemma 1 (Push-forward Posterior from SCM to Distributions).** *Let $D_n$ denote the $n$ in-context examples (i.e., evidence) for PFN to inference, where $D_n$ might contain both interventional ($\tilde{X}$) or observational ($X$) samples. Moreover, we define the likelihood of $D_n$ under an SCM $M \in \mathcal{M}$ by $L(D_n \mid M)$, and $L(D_n \mid P)$ for each $P \in \mathcal{P}$. Let $\Lambda$ be any prior on $\mathcal{M}$, and let $\Pi := \Phi_{\#}\Lambda$ be its push-forward prior on $\mathcal{P}$, i.e. $\Pi(B) = \Lambda(\Phi^{-1}(B))$ for any $B \subseteq \mathcal{P}$. Define the corresponding posteriors:*

$$\Lambda_n(A) := \frac{\int_A L(D_n \mid M)\, \Lambda(dM)}{\int_{\mathcal{M}} L(D_n \mid M)\, \Lambda(dM)}, \qquad \Pi_n(B) := \frac{\int_B L(D_n \mid P)\, \Pi(dP)}{\int_{\mathcal{P}} L(D_n \mid P)\, \Pi(dP)}.$$

*Then the posteriors satisfy $\Phi_{\#}\Lambda_n = \Pi_n$, i.e., for all measurable $B \subseteq \mathcal{P}$, $\Lambda\big(\Phi^{-1}(B) \mid D_n\big) = \Pi(B \mid D_n)$.*

*Proof.* We re-write the in-context samples as $D_n = \{(a^{(i)}, x^{(i)})\}_{i=1}^n$, the label $a^{(i)}$ denotes the experimental condition under which $x^{(i)}$ is drawn. This condition may correspond to:

- The observational environment, $a^{(i)} = \text{obs}$, in which case $x^{(i)} \sim P_M^{\text{obs}} = P_M$, or
- An interventional environment, $a^{(i)} = \text{do}(\tilde{X} = \tilde{x})$, in which case $x^{(i)} \sim P_M^{\text{do}(\tilde{X}=\tilde{x})}$.

Thus the likelihood under an SCM $M \in \mathcal{M}$ is

$$L(D_n \mid M) = \prod_{i=1}^n P_M^{a^{(i)}}(x^{(i)}).$$

By construction of $\Phi$, the image $\Phi(M)$ is not a single marginal law but the full family $\{P_M^a : a \in \mathcal{A}\}$ of observational and interventional distributions induced by $M$. Hence, as any in-context samples $D_n = \{(a^{(i)}, x^{(i)})\}_{i=1}^n$ are sampled from the family $\{P_M^a\}_a$, the joint likelihood $\prod_{i=1}^n P_M^{a^{(i)}}(x^{(i)})$ admits the following equality:

$$L(D_n \mid M) = \prod_{i=1}^n P_{\Phi(M)}^{a^{(i)}}(x^{(i)}) = L(D_n \mid \Phi(M)). \tag{$*$}$$

Now fix any measurable $B \subseteq \mathcal{P}$. Using Bayes' rule,

$$\Phi_{\#}\Lambda_n(B) = \Lambda_n(\Phi^{-1}(B)) = \Lambda(\Phi^{-1}(B) \mid D_n) = \frac{\displaystyle\int_{\Phi^{-1}(B)} L(D_n \mid M)\, \Lambda(dM)}{\displaystyle\int_{\mathcal{M}} L(D_n \mid M)\, \Lambda(dM)}.$$

Applying $(*)$ yields

$$\Lambda(\Phi^{-1}(B) \mid D_n) = \frac{\displaystyle\int_{\Phi^{-1}(B)} L(D_n \mid \Phi(M))\, \Lambda(dM)}{\displaystyle\int_{\mathcal{M}} L(D_n \mid \Phi(M))\, \Lambda(dM)}.$$

Since $\Pi = \Phi_{\#}\Lambda$, for any measurable $f : \mathcal{P} \to \mathbb{R}$,

$$\int_{\mathcal{M}} f(\Phi(M))\, \Lambda(dM) = \int_{\mathcal{P}} f(P)\, \Pi(dP).$$

Choosing $f(P) = L(D_n \mid P)\mathbf{1}_B(P)$ and $f(P) = L(D_n \mid P)$ respectively gives

$$\int_{\Phi^{-1}(B)} L(D_n \mid \Phi(M))\, \Lambda(dM) = \int_B L(D_n \mid P)\, \Pi(dP),$$

$$\int_{\mathcal{M}} L(D_n \mid \Phi(M))\, \Lambda(dM) = \int_{\mathcal{P}} L(D_n \mid P)\, \Pi(dP).$$

Substituting these identities into the expression for $\Lambda(\Phi^{-1}(B) \mid D_n)$ yields

$$\Lambda(\Phi^{-1}(B) \mid D_n) = \frac{\int_B L(D_n \mid P)\, \Pi(dP)}{\int_{\mathcal{P}} L(D_n \mid P)\, \Pi(dP)} = \Pi(B \mid D_n) = \Pi_n(B).$$

Thus $\Phi_{\#}\Lambda_n = \Pi_n$, completing the proof. $\qquad\square$

## A.2. Theory of Uncovered Probability

**Theorem 1 [Exponential Prior Uncoverage under Finite Interventional Pretraining]** *Let $\Pi$ be a meta-prior over $S_{\text{int}}$ such that $\pi(\tilde{X}, \tilde{x}) > 0$ for all $(\tilde{X}, \tilde{x})$. Suppose a PFN is pre-trained on a finite subset $\overline{S}_{\text{int}} = \{P^{(1)}, \dots, P^{(N)}\} \subset S_{\text{int}}$, yielding the empirical prior $\hat{\Pi}$. Then the uncovered prior mass*

$$\Pi\big(S^p_{\text{zero}}\big) \;=\; \Pi\big(S_{\text{int}} \setminus \overline{S}_{\text{int}}\big)$$

*satisfies*

$$\Pi\big(S^p_{\text{zero}}\big) \;\geq\; 1 - \frac{N}{(1+d)^D},$$

*and in particular, for any polynomially bounded $N = \text{poly}(D, d)$,*

$$\lim_{D \to \infty} \Pi\big(S^p_{\text{zero}}\big) \;=\; 1,$$

*i.e., the uncovered interventional prior mass converges to one exponentially fast in the number of variables $D$.*

*Proof.* We first lower bound the cardinality of the full interventional set $S_{\text{int}}$. For each variable $X_i$, there are two possibilities: either $X_i$ is not intervened, or it is intervened and fixed to a value in $\mathcal{X}_i$. Since $|\mathcal{X}_i| \leq d$, each variable admits at most $(1+d)$ distinct intervention states. Therefore, the total number of distinct hard interventions satisfies

$$|S_{\text{int}}| \;\geq\; \prod_{i=1}^{D}(1 + |\mathcal{X}_i|) \;\geq\; (1+d)^D.$$

Since the meta-prior $\Pi$ assigns strictly positive probability mass to each interventional configuration, the maximum total prior mass that can be covered by $N$ distinct interventional distributions is at most $N/(1+d)^D$. Consequently, the uncovered prior mass satisfies

$$\Pi\big(S^p_{\text{zero}}\big) \;=\; 1 - \Pi(\overline{S}_{\text{int}}) \;\geq\; 1 - \frac{N}{(1+d)^D}.$$

Finally, when $N$ grows at most polynomially in $(D, d)$, the ratio $N/(1+d)^D$ decays exponentially fast to zero as $D \to \infty$, implying that $\Pi(S^p_{\text{zero}}) \to 1$. $\qquad\square$

## A.3. General Lower Divergence Bound in Prior Space

**Theorem 1. (Prior Divergence Lower Bound under Non-Uniform Prior)** *Let the total mass of $S^p_{\text{zero}}$ under the grounding prior $\Pi$ as*

$$\delta := \Pi(S^p_{\text{zero}}) = \sum_{P^{\text{do}(\tilde{X}=\tilde{x})} \in S^p_{\text{zero}}} \pi(\tilde{X}, \tilde{x}).$$

*Suppose further that any uncovered interventional distribution $P^{\text{do}(\tilde{X}=\tilde{x})} \in S^p_{\text{zero}}$ has total-variation distance to the nearest covered one satisfying*

$$\inf_{Q \in \overline{S}_{\text{int}}} \text{TV}\Big(P^{\text{do}(\tilde{X}=\tilde{x})}, Q\Big) \geq \varepsilon_0 > 0.$$

*Then, the following universal lower bounds hold:*

$$\boxed{\text{TV}(\Pi, \hat{\Pi}) \geq \delta, \qquad W_{\text{TV}}(\Pi, \hat{\Pi}) \geq \varepsilon_0\, \delta.} \tag{17}$$

*Proof.* From the definition of total variation distance between two distributions over $\mathcal{S}_{\text{int}}$,

$$\text{TV}(\Pi, \hat{\Pi}) = \sup_{A \subseteq \mathcal{S}_{\text{int}}} \left| \Pi(A) - \hat{\Pi}(A) \right|.$$

By taking $A = S^p_{\text{zero}}$, we have $\hat{\Pi}(A) = 0$ and $\Pi(A) = \delta$, hence $\text{TV}(\Pi, \hat{\Pi}) \geq \delta$. For the transportation cost under the TV metric,

$$W_{\text{TV}}(\Pi, \hat{\Pi}) = \inf_{\gamma \in \Gamma(\Pi, \hat{\Pi})} \int \text{TV}(P, Q) \, d\gamma(P, Q),$$

where $\Gamma(\Pi, \hat{\Pi})$ denotes the set of all valid couplings. Since all prior mass $\delta$ over $S^p_{\text{zero}}$ must be transported to the empirical support $\overline{S}_{\text{int}}$, and each such transport incurs a cost of at least $\varepsilon_0$ by assumption, we obtain the lower bound

$$W_{\text{TV}}(\Pi, \hat{\Pi}) \geq \varepsilon_0 \, \delta.$$

$\square$

## A.4. Examples: Prior Divergence Bound on Priors

**Corollary 7** (Concrete priors: three cases)**.** *Under the notation and assumptions of Theorem 2, assume the pretrained model observes $N$ interventional distributions $\overline{S}_{\text{int}} = \{P^{(i)}\}_{i=1}^N$ and let $\varepsilon_0 > 0$ be the minimal TV gap from uncovered to covered distributions as in the theorem. Then the following hold for the three concrete choices of the true meta-prior $\Pi$:*

1. ***Uniform prior over interventions.***
   *If*
   $$\pi(\tilde{X}, \tilde{x}) = \frac{1}{(1+d)^D} \qquad \text{for all } \tilde{X} \subseteq X, \, \tilde{x} \in \mathcal{X}_{\tilde{X}},$$
   *then the uncovered mass equals*
   $$\delta = 1 - \frac{N}{(1+d)^D},$$
   *and thus*
   $$\text{TV}(\Pi, \hat{\Pi}) \geq 1 - \frac{N}{(1+d)^D}, \qquad W_{\text{TV}}(\Pi, \hat{\Pi}) \geq \varepsilon_0 \left(1 - \frac{N}{(1+d)^D}\right).$$

2. ***Size-penalized prior (scale $\lambda > 0$).***
   *Suppose the prior penalizes interventions by cardinality $|\tilde{X}|$ via*
   $$\pi(\tilde{X}, \tilde{x}) = \frac{\lambda^{|\tilde{X}|}}{Z_D} \cdot \frac{1}{d^{|\tilde{X}|}}, \qquad Z_D := \sum_{k=0}^{D} \binom{D}{k} \lambda^k,$$
   *i.e. all assignments of the same intervened set are equally likely under the conditional of that set. Then the uncovered mass is*
   $$\delta = 1 - \frac{1}{Z_D} \sum_{P^{(i)} \in \overline{S}_{\text{int}}} \lambda^{|\tilde{X}^{(i)}|},$$
   *where $|\tilde{X}^{(i)}|$ is the number of variables intervened in $P^{(i)}$. Hence*
   $$\text{TV}(\Pi, \hat{\Pi}) \geq 1 - \frac{1}{Z_D} \sum_{P^{(i)} \in \overline{S}_{\text{int}}} \lambda^{|\tilde{X}^{(i)}|}, \quad W_{\text{TV}}(\Pi, \hat{\Pi}) \geq \varepsilon_0 \left(1 - \frac{1}{Z_D} \sum_{P^{(i)} \in \overline{S}_{\text{int}}} \lambda^{|\tilde{X}^{(i)}|}\right).$$

3. ***Graph-structured prior (structure bias $\alpha \geq 0$).***
   *Let $\deg_G(\tilde{X})$ denote a graph-derived complexity of the intervened set (e.g. total degree in $G$), and define*
   $$\pi(\tilde{X}, \tilde{x}) = \frac{\exp\left(-\alpha \deg_G(\tilde{X})\right)}{Z_G} \cdot \frac{1}{d^{|\tilde{X}|}}, \qquad Z_G := \sum_{\tilde{X} \subseteq X} \binom{d^{|\tilde{X}|}}{1} \exp\left(-\alpha \deg_G(\tilde{X})\right).$$

*(The factor $1/d^{|\tilde{X}|}$ distributes mass uniformly over assignments $\tilde{x}$ for fixed $\tilde{X}$; $Z_G$ normalizes over all intervened sets and assignments.) Then the uncovered mass is*

$$\delta \;=\; 1 - \frac{1}{Z_G} \sum_{P^{(i)} \in \overline{S}_{\mathrm{int}}} \exp\big(-\alpha \deg_G(\tilde{X}^{(i)})\big),$$

*and consequently*

$$\mathrm{TV}(\Pi, \hat{\Pi}) \geq 1 - \frac{1}{Z_G} \sum_{P^{(i)} \in \overline{S}_{\mathrm{int}}} \exp\big(-\alpha \deg_G(\tilde{X}^{(i)})\big),$$

$$W_{\mathrm{TV}}(\Pi, \hat{\Pi}) \geq \varepsilon_0 \Big( 1 - \frac{1}{Z_G} \sum_{P^{(i)} \in \overline{S}_{\mathrm{int}}} \exp\big(-\alpha \deg_G(\tilde{X}^{(i)})\big) \Big).$$

*Proof.* Each case is a direct instantiation of Theorem 2. Compute the uncovered prior mass $\delta = \Pi(\mathcal{S}_{\mathrm{int}} \setminus \overline{S}_{\mathrm{int}}) = 1 - \sum_{P^{(i)} \in \overline{S}_{\mathrm{int}}} \pi(\tilde{X}^{(i)}, \tilde{x}^{(i)})$ under the specified $\pi(\cdot)$, then apply the bounds $\mathrm{TV}(\Pi, \hat{\Pi}) \geq \delta$ and $W_{\mathrm{TV}}(\Pi, \hat{\Pi}) \geq \varepsilon_0 \delta$. $\qquad\square$

**Remark 1 (Uniform prior).** When $\Pi$ is uniform, all possible interventions are equally likely. The uncovered mass $\delta = 1 - \frac{N}{(1+d)^D}$ grows exponentially with $M$, indicating that the empirical prior $\hat{\Pi}$ quickly loses support as the causal system dimension increases. Hence, even large-scale pretraining cannot achieve full coverage, making zero-shot inference over unseen interventions theoretically impossible.

**Remark 2 (Size-penalized prior).** This prior favors smaller intervention sets through the hyperparameter $\lambda < 1$. As $\lambda$ decreases, $\Pi$ concentrates on low-order interventions, reducing $\delta$ for those but enlarging the uncovered mass for large-scale ones. Consequently, PFN pretraining may achieve few-shot generalization for single-variable interventions, yet still fails on multi-variable (combinatorial) interventions—an implicit form of the exponential blow-up problem in causal coverage.

**Remark 3 (Graph-structured prior).** Here the prior encodes structural inductive bias from the causal graph $G$ via $\deg_G(\tilde{X})$. When $\alpha > 0$, interventions on highly connected nodes receive smaller prior mass, focusing learning on local interventions. However, if pretraining lacks exposure to high-degree variables, the uncovered prior mass $\delta$ remains significant, yielding a lower bound on distributional divergence even under graph-aware meta-priors. This explains why PFN-style models with limited structure coverage cannot guarantee consistent posterior inference across all causal mechanisms.

## A.5. Inconsistent Estimations of Interventional Query

**Theorem 2. (Posterior inconsistency under prior divergence)** *Let $\Pi$ denote the true meta-prior over interventional distributions $\{P_{\mathrm{int}}\}$, and $\hat{\Pi}$ the empirical prior induced by $N$ observed interventions $\overline{S}_{\mathrm{int}} = \{P_{\mathrm{int}}^{(i)}\}_{i=1}^{N}$. For any functional of the interventional distribution*

$$Q(P_{\mathrm{int}}) = \int q(x)\, P_{\mathrm{int}}(dx), \tag{18}$$

*let $\Pi_N(Q)$ denote the posterior of $Q$ under $\hat{\Pi}$ after observing $N$ samples from the causal system. Assuming that:*
*(1) the regularity conditions of the Bernstein–von Mises theorem hold for the true prior $\Pi$: the likelihood is sufficiently smooth, $Q$ is a differentiable functional of $P_{\mathrm{int}}$, and the Fisher information $I(Q_*)$ at the true value $Q_*$ is positive definite;*
*(2) The queried interventional statistics distinguish distributions in $S_{\mathrm{zero}}^{p}$ from its complement, i.e., there exists a constant $\eta > 0$ such that*

$$\inf_{P \in S_{\mathrm{zero}}^{p}} \inf_{P' \in \overline{S}_{\mathrm{int}}} \big|Q(P) - Q(P')\big| \;\geq\; \eta \|P - P'\|_{TV}. \tag{19}$$

*(3) The function $q : \mathcal{X} \to \mathbb{R}$ is measurable and bounded:*

$$\|q\|_{\infty} := \sup_{x \in \mathcal{X}} |q(x)| < \infty.$$

*When conditions in Theorem 2 holds, then the interventional query induced by $\hat{\Pi}$ admits the lower bound on the posterior bias:*

$$\mathbb{E}_{P_{\mathrm{int}} \sim \hat{\Pi}}\big[|Q(P_{\mathrm{int}}) - Q_*|\big] \;\geq\; \eta\delta - O(n^{-1/2}).$$

*Moreover, the posterior contraction rate satisfies*

$$\|\Pi_N(Q) - \mathcal{N}(Q_*, I^{-1}(Q_*)/N)\|_{\mathrm{TV}} \geq \delta - O(n^{-1/2}),$$

*which implies that the posterior cannot asymptotically converge to the nominal Gaussian limit unless $N \gg \delta^{-2}$ or $\mathrm{TV}(\Pi, \hat{\Pi}) \to 0$.*

*Proof of Corollary 3.* Let $\Pi$ denote the true meta-prior over interventional distributions $\{P_{\mathrm{int}}\}$ and $\hat{\Pi}$ the empirical prior induced by $N$ observed interventional components $\overline{S}_{\mathrm{int}} = \{P_{\mathrm{int}}^{(i)}\}_{i=1}^N$. Let $A = S_{\mathrm{zero}}^p$ be the subset of unseen interventional distributions such that $\Pi(A) = \delta > 0$ and $\hat{\Pi}(A) = 0$. Denote $A^c$ its complement, the set of covered interventions with $\hat{\Pi}(A^c) = 1$.

**Step 1. Prior decomposition.** The true prior $\Pi$ can be written as a convex mixture

$$\Pi = (1 - \delta)\,\Pi_{A^c} + \delta\,\Pi_A,$$

where $\Pi_{A^c}(\cdot) = \Pi(\cdot \mid A^c)$ and $\Pi_A(\cdot) = \Pi(\cdot \mid A)$ are the conditional priors on the covered and uncovered regions. Since the empirical prior $\hat{\Pi}$ is only supported on $A^c$, it can be expressed as $\hat{\Pi} = (1 - \delta)\tilde{\Pi}_{A^c}$ for some probability measure $\tilde{\Pi}_{A^c}$ supported on $A^c$. Hence, the discrepancy between the true and empirical priors is

$$\mathrm{TV}(\Pi, \hat{\Pi}) = \delta + (1 - \delta)\mathrm{TV}(\Pi_{A^c}, \tilde{\Pi}_{A^c}) \geq \delta.$$

**Step 2. Derivation of the conditional posteriors.** We start from Bayes' rule for the (unnormalized) posterior measure over the space of interventional distributions $S_{\mathrm{int}}$:

$$d\Pi_n^{(\Pi)}(P) \;\propto\; L_n(P)\,d\Pi(P),$$

where $L_n(P)$ denotes the marginal likelihood of the observed data under model $P$, and $\Pi$ is the prior over $S_{\mathrm{int}}$.

Partition the parameter space into two disjoint measurable sets $A$ (uncovered interventions) and $A^c$ (covered interventions). For any measurable test set $B \subseteq S_{\mathrm{int}}$ we may write

$$\int_B L_n(P)\,d\Pi(P) = \int_{B \cap A^c} L_n(P)\,d\Pi(P) + \int_{B \cap A} L_n(P)\,d\Pi(P).$$

Define the (prior) conditional measures on $A^c$ and $A$:

$$\Pi_{A^c}(C) \;=\; \frac{\Pi(C \cap A^c)}{\Pi(A^c)}, \qquad \Pi_A(C) \;=\; \frac{\Pi(C \cap A)}{\Pi(A)},$$

for any measurable $C \subseteq S_{\mathrm{int}}$, provided $\Pi(A^c) > 0$ and $\Pi(A) > 0$. (When a denominator is zero the corresponding conditional measure is undefined; one then treats the expressions in the limiting or trivial sense.)

Using these conditional priors we can rewrite the integrals appearing in the posterior as

$$\int_{B \cap A^c} L_n(P)\,d\Pi(P) = \Pi(A^c) \int_{B \cap A^c} L_n(P)\,d\Pi_{A^c}(P),$$

and

$$\int_{B \cap A} L_n(P)\,d\Pi(P) = \Pi(A) \int_{B \cap A} L_n(P)\,d\Pi_A(P).$$

Now the posterior probability of $B$ is the normalized version of the unnormalized integral:

$$\Pi_n^{(\Pi)}(B) = \frac{\displaystyle\int_B L_n(P)\,d\Pi(P)}{\displaystyle\int_{S_{\mathrm{int}}} L_n(P)\,d\Pi(P)} = \frac{\displaystyle\int_{B \cap A^c} L_n(P)\,d\Pi(P) + \int_{B \cap A} L_n(P)\,d\Pi(P)}{\displaystyle\int_{A^c} L_n(P)\,d\Pi(P) + \int_A L_n(P)\,d\Pi(P)}.$$

**Remark in Proof [1].** *The above derivations follow the typical Bayesian formula, with the following detailed expansions. Given $n$ observations with marginal likelihood $L_n(P)$, the posterior measure $\Pi_n^{(\Pi)}$ is defined by Bayes' rule as:*

$$d\Pi_n^{(\Pi)}(P) \;=\; \frac{L_n(P)\,d\Pi(P)}{\displaystyle\int_{\mathcal{S}_{\text{int}}} L_n(P')\,d\Pi(P')}. \tag{20}$$

*For any measurable subset $B \subseteq \mathcal{S}_{\text{int}}$, the posterior probability of $B$ is then*

$$\Pi_n^{(\Pi)}(B) \;=\; \int_B d\Pi_n^{(\Pi)}(P) \;=\; \frac{\displaystyle\int_B L_n(P)\,d\Pi(P)}{\displaystyle\int_{\mathcal{S}_{\text{int}}} L_n(P)\,d\Pi(P)}. \tag{21}$$

Restricting to the case where we condition on $A^c$ (i.e. consider the posterior mass of $B$ relative to the event $A^c$), we obtain the conditional posterior on $A^c$ by renormalizing the posterior restricted to $A^c$:

$$\Pi_n^{(A^c)}(B) := \Pi_n^{(\Pi)}(B \mid A^c) = \frac{\Pi_n^{(\Pi)}(B \cap A^c)}{\Pi_n^{(\Pi)}(A^c)} = \frac{\displaystyle\int_{B\cap A^c} L_n(P)\,d\Pi(P)}{\displaystyle\int_{A^c} L_n(P)\,d\Pi(P)}.$$

The algebraic steps used are just Bayes' rule plus the definition of conditional probability, and the denominator is the posterior mass of $A^c$ (assumed positive).

**Step 3. Posterior decomposition.** Let $A = S_{\text{zero}}^p$ be the set of uncovered interventional distributions and $A^c = S_{\text{int}} \setminus A$ its complement. For any measurable subset $B \subseteq S_{\text{int}}$ (here $B$ is an arbitrary measurable event in the space of interventional distributions, e.g. $B = \{P\}$ or $B = A$), we denote by $L_n(P)$ the marginal likelihood of the observed data under model $P$.

Step 2 provides conditional (restricted-and-renormalized) posteriors on $A^c$ and $A$ by

$$\Pi_n^{(A^c)}(B) \;=\; \frac{\displaystyle\int_{B\cap A^c} L_n(P)\,d\Pi(P)}{\displaystyle\int_{A^c} L_n(P)\,d\Pi(P)}, \qquad \Pi_n^{(A)}(B) \;=\; \frac{\displaystyle\int_{B\cap A} L_n(P)\,d\Pi(P)}{\displaystyle\int_{A} L_n(P)\,d\Pi(P)}.$$

(These are well-defined probability measures provided the denominators are positive; if a denominator is zero the corresponding conditional posterior is degenerate and the following algebra should be interpreted in the limiting sense.)

Using the prior decomposition in Step 1 as

$$\Pi = (1-\delta)\,\Pi_{A^c} + \delta\,\Pi_A, \qquad \hat{\Pi} = (1-\delta)\,\tilde{\Pi}_{A^c},$$

the unnormalized posterior under $\Pi$ has total mass (normalizing constant)

$$Z_n \;=\; \int L_n(P)\,d\Pi(P) \;=\; (1-\delta)\int_{A^c} L_n(P)\,d\Pi_{A^c}(P) + \delta \int_A L_n(P)\,d\Pi_A(P).$$

Set

$$Z_n^{(A^c)} := \int_{A^c} L_n(P)\,d\Pi_{A^c}(P), \qquad Z_n^{(A)} := \int_A L_n(P)\,d\Pi_A(P),$$

so that $Z_n = (1-\delta)Z_n^{(A^c)} + \delta Z_n^{(A)}$. Consequently, the (normalized) posterior under $\Pi$ can be written as the convex mixture

$$\Pi_n^{(\Pi)} \;=\; w_n\,\Pi_n^{(A^c)} + (1-w_n)\,\Pi_n^{(A)}, \tag{22}$$

where the mixture weight $w_n$ is exactly

$$w_n \;=\; \frac{(1-\delta)Z_n^{(A^c)}}{(1-\delta)Z_n^{(A^c)} + \delta Z_n^{(A)}} \;=\; \frac{(1-\delta)Z_n^{(A^c)}}{Z_n} \in (0,1). \tag{23}$$

Intuitively, $w_n$ is the posterior probability (under $\Pi$) assigned to the covered region $A^c$.

By contrast, the empirical prior $\hat{\Pi}$ is supported on $A^c$, hence its posterior is the renormalized restriction of the likelihood to $A^c$ with respect to $\tilde{\Pi}_{A^c}$:

$$\Pi_n^{(\hat{\Pi})}(B) \;=\; \frac{\int_{B \cap A^c} L_n(P)\, d\tilde{\Pi}_{A^c}(P)}{\int_{A^c} L_n(P)\, d\tilde{\Pi}_{A^c}(P)} \;=:\; \tilde{\Pi}_n^{(A^c)}(B), \tag{24}$$

so $\Pi_n^{(\hat{\Pi})}$ is supported entirely on $A^c$.

Moreover, observe that $\Pi_n^{(\Pi)}$ assigns mass $(1 - w_n)$ to $A$ while $\Pi_n^{(\hat{\Pi})}$ assigns mass $0$ to $A$. Therefore the total-variation distance between the two posteriors satisfies the immediate lower bound

$$\mathrm{TV}\big(\Pi_n^{(\Pi)}, \Pi_n^{(\hat{\Pi})}\big) \;\geq\; \big|\Pi_n^{(\Pi)}(A) - \Pi_n^{(\hat{\Pi})}(A)\big| \;=\; 1 - w_n. \tag{25}$$

(Indeed TV distance is at least the absolute mass difference on any measurable set; here take the set $A$.)

**Remark in Proof [2]:** $w_n \to 1 - \delta$ **with infinite** $n$. *To be first, the marginal likelihood can be written as*

$$L_n(P) \approx \exp\{-n D_{\mathrm{KL}}\left(P^\star \| P\right)\}, \tag{26}$$

*and the integral $Z_n^{(A)} := \int_A L_n(P)\, d\Pi_A(P)$ will tend to zero as the region $A$ does not contain the true distribution corresponding to the interventional query $P_{\mathrm{int}}^*$. Then together with the regularity conditions, i.e., namely that the likelihood $L_n(P)$ concentrates near the true data-generating distribution $P_{\mathrm{int}}^*$, the true prior $\Pi$ assigns positive mass to $P_{\mathrm{int}}^*$, and the normalizing integrals $Z_n^{(A)}$ and $Z_n^{(A^c)}$ remains regular, the uncovered component's contribution vanishes asymptotically:*

$$\frac{Z_n^{(A)}}{Z_n^{(A^c)}} \to 0, \quad w_n \to \frac{1 - \delta}{(1 - \delta) + 0} = 1 - \delta \tag{27}$$

**Step 3. Bounding posterior discrepancy.** Let $\mathcal{N}_n := \mathcal{N}(Q_*, I^{-1}/n)$ denote the Gaussian limit appearing in the Bernstein–von Mises theorem for the posterior of $Q$ under the true prior $\Pi$. By the triangle inequality (applied with $a = \Pi_n^{(\hat{\Pi})}$, $b = \Pi_n^{(\Pi)}$, $c = \mathcal{N}_n$),

$$\|\Pi_n^{(\hat{\Pi})} - \mathcal{N}_n\|_{\mathrm{TV}} \;\geq\; \|\Pi_n^{(\hat{\Pi})} - \Pi_n^{(\Pi)}\|_{\mathrm{TV}} - \|\Pi_n^{(\Pi)} - \mathcal{N}_n\|_{\mathrm{TV}}.$$

Using (25) and the Bernstein–von Mises convergence $\|\Pi_n^{(\Pi)} - \mathcal{N}_n\|_{\mathrm{TV}} = O(^{-1/2})$, we obtain

$$\boxed{\|\Pi_n^{(\hat{\Pi})} - \mathcal{N}_n\|_{\mathrm{TV}} \;\geq\; (1 - w_n) - O(^{-1/2}).}$$

Taking the $\liminf$ as $n \to \infty$ and recalling that under regular likelihood concentration $w_n \to 1 - \delta$ (so $1 - w_n \to \delta$), we conclude

$$\boxed{\liminf_{n \to \infty} \|\Pi_n^{(\hat{\Pi})} - \mathcal{N}_n\|_{\mathrm{TV}} \;\geq\; \delta,}$$

and for finite $n$ the non-asymptotic bound

$$\boxed{\|\Pi_n^{(\hat{\Pi})} - \mathcal{N}_n\|_{\mathrm{TV}} \geq \delta - O(^{-1/2}),}$$

holds. Equivalently, even with enough posterior evidence, i.e., $n \gg \delta^{-2}$ (so that $O(^{-1/2}) \ll \delta$), the gap as $\delta$ still remains between the estimated posterior under the empirical prior $\hat{\Pi}$ and the BvM Gaussian limit $\mathcal{N}_n$.

**Step 4. Posterior Interventional Query.**

Let $Q(P_{\mathrm{int}}) = \int q(x)\, dP_{\mathrm{int}}(x)$ be the interventional query of interest, with $\|q\|_\infty < \infty$. Denote the true causal value $Q_* := Q(P_{\mathrm{int}}^*)$. We consider the posterior under the empirical meta-prior $\hat{\Pi}$ and ask how prior-level discrepancies manifest in the posterior expectation $\mathbb{E}_{\Pi_N^{(\hat{\Pi})}}[Q]$.

**Truth uncovered by $\hat{\Pi}$ (misspecified / irrecoverable bias).** Assume that the true interventional distribution $P_{\text{int}}^\star$ is not contained in the support of $\hat{\Pi}$, i.e. $P_{\text{int}}^\star \notin \text{supp}(\hat{\Pi})$, while it lies within the support of the grounding prior $\Pi$. Under standard regularity assumptions for the Bernstein–von Mises theorem (differentiable likelihood, positive-definite Fisher information, identifiable model), the posterior under the true prior $\Pi$ admits the Gaussian approximation

$$\Pi_N^{(\Pi)}(\,P_{\text{int}}\,) \approx \mathcal{N}\big(P_{\text{int}}^\star, I(P_{\text{int}}^\star)^{-1}/N\big),$$

Applying the *functional delta method* to a smooth functional $Q : \mathcal{P} \to \mathbb{R}$ that is Fréchet differentiable at $P_{\text{int}}^\star$ with influence function $\phi_Q(x) = \frac{Q(P_{\text{int}})}{P_{\text{int}}}\big|_{P_{\text{int}} = P_{\text{int}}^\star}$ (Castillo & Rousseau, 2015; Rivoirard & Rousseau, 2012), we obtain

$$\sqrt{N}\big(Q(P_{\text{int}}) - Q_*\big) \Rightarrow \mathcal{N}\big(0, \, \nabla Q(P_{\text{int}}^\star)^\top I(P_{\text{int}}^\star)^{-1} \nabla Q(P_{\text{int}}^\star)\big),$$

and thus

$$\mathbb{E}_{\Pi_N^{(\Pi)}}[Q] = Q_* + O(^{-1/2}).$$

Intuitively, this shows that posterior fluctuations of $Q$ scale as $^{-1/2}$ around its true causal value $Q_*$, provided the prior covers $P_{\text{int}}^\star$. Therefore the posterior query bias satisfies the exact relation

$$
\begin{aligned}
\big|\mathbb{E}_{\Pi_N^{(\hat{\Pi})}}[Q] - Q_*\big| &= \big|\mathbb{E}_{\Pi_N^{(\hat{\Pi})}}[Q] - \mathbb{E}_{\Pi_N^{(\Pi)}}[Q] + \mathbb{E}_{\Pi_N^{(\Pi)}}[Q] - Q_*\big| \\
&\geq \big|\mathbb{E}_{\Pi_N^{(\hat{\Pi})}}[Q] - \mathbb{E}_{\Pi_N^{(\Pi)}}[Q]\big| - \big|\mathbb{E}_{\Pi_N^{(\Pi)}}[Q] - Q_*\big| \\
&\geq \eta\left(\delta - O(^{-1/2})\right) - O(^{-1/2}) \\
&= \eta\delta - O(^{-1/2}).
\end{aligned}
$$

This proves the theorem. $\qquad\qquad\qquad\qquad\qquad\qquad\qquad\qquad\qquad\qquad\qquad\qquad\qquad\qquad$ $\square$

**Remark.** This corollary formalizes how a non-vanishing prior gap $(\text{TV}(\Pi, \hat{\Pi}) \geq \delta)$ propagates through the Bernstein–von Mises mechanism: the posterior cannot collapse to the correct Gaussian asymptotic form unless the number of observations grows as $N \gg \delta^{-2}$. Intuitively, even if the data likelihood is highly informative, the posterior remains biased towards regions unsupported by the empirical prior. This explains why in causal meta-pretraining, incomplete intervention coverage produces systematic posterior bias and prevents zero-shot recovery of unseen causal quantities $Q(P_{\text{int}})$.

**Corollary 8** (Posterior TV gap lifts from $\mathcal{P}$ to $\mathcal{M}$). *Let $\Lambda, \hat{\Lambda}$ be two priors on $\mathcal{M}$, and $\Pi = \Phi_\#\Lambda, \hat{\Pi} = \Phi_\#\hat{\Lambda}$ their induced priors on $\mathcal{P}$. Let $\Lambda_n, \hat{\Lambda}_n$ and $\Pi_n, \hat{\Pi}_n$ be the respective posteriors given $D_n$. Then*

$$\|\Pi_n - \hat{\Pi}_n\|_{\text{TV}} = \|\Phi_\#\Lambda_n - \Phi_\#\hat{\Lambda}_n\|_{\text{TV}} \leq \|\Lambda_n - \hat{\Lambda}_n\|_{\text{TV}}.$$

*In particular, if for some sequence $n$,*

$$\|\Pi_n - \hat{\Pi}_n\|_{\text{TV}} \geq \delta - O(n^{-1/2}) \quad \Rightarrow \quad \|\Lambda_n - \hat{\Lambda}_n\|_{\text{TV}} \geq \delta - O(n^{-1/2}).$$

*Proof.* By Lemma 1 we have the pushforward identities

$$\Pi_n = \Phi_\#\Lambda_n, \qquad \hat{\Pi}_n = \Phi_\#\hat{\Lambda}_n.$$

Recall the total variation distance between two probability measures $\mu, \nu$ on a measurable space $(\mathcal{X}, \mathcal{B})$ can be written as

$$\|\mu - \nu\|_{\text{TV}} := \sup_{A \in \mathcal{B}} \big|\mu(A) - \nu(A)\big| = \tfrac{1}{2}\int_{\mathcal{X}} \big|d\mu - d\nu\big|.$$

Let $\Phi : (\mathcal{M}, \mathcal{B}_{\mathcal{M}}) \to (\mathcal{P}, \mathcal{B}_{\mathcal{P}})$ be measurable (Lemma 1 has informed that $\Phi$ is measure) and let $\Phi_\#$ denote the pushforward operator. For any measurable set $B \in \mathcal{B}_{\mathcal{P}}$ we have by the definition of pushforward

$$\Phi_\#\mu(B) = \mu\big(\Phi^{-1}(B)\big), \qquad \Phi_\#\nu(B) = \nu\big(\Phi^{-1}(B)\big).$$

Hence

$$\left|\Phi_{\#}\mu(B) - \Phi_{\#}\nu(B)\right| = \left|\mu(\Phi^{-1}(B)) - \nu(\Phi^{-1}(B))\right| \leq \sup_{A \in \mathcal{B}_{\mathcal{M}}} \left|\mu(A) - \nu(A)\right| = \|\mu - \nu\|_{\text{TV}}.$$

Taking the supremum over all measurable $B \subseteq \mathcal{P}$ yields the non-expansiveness property of pushforward in total variation:

$$\|\Phi_{\#}\mu - \Phi_{\#}\nu\|_{\text{TV}} := \sup_{B \in \mathcal{B}_{\mathcal{P}}} \left|\Phi_{\#}\mu(B) - \Phi_{\#}\nu(B)\right| \leq \|\mu - \nu\|_{\text{TV}}.$$

Applying this inequality with $\mu = \Lambda_n$ and $\nu = \hat{\Lambda}_n$, and using the identities $\Pi_n = \Phi_{\#}\Lambda_n$ and $\hat{\Pi}_n = \Phi_{\#}\hat{\Lambda}_n$, we obtain

$$\|\Pi_n - \hat{\Pi}_n\|_{\text{TV}} = \|\Phi_{\#}\Lambda_n - \Phi_{\#}\hat{\Lambda}_n\|_{\text{TV}} \leq \|\Lambda_n - \hat{\Lambda}_n\|_{\text{TV}},$$

which completes the proof. $\square$

# B. Proof of Theories: Interventional fine-tuning of PFNs

## B.1. Generalizations of PWF and MSF

**Theorem 5. (Robustness of PFN under TV-ball perturbations)** *Let $P_{\text{int}}^0$ denote the point-wise interventional distribution for fine-tuning, and let $\mathcal{B}_{\varepsilon}(P_{\text{int}}^0) := \{P : \text{TV}(P_W, P_{\text{int}}^{0,W}) \leq \varepsilon\}$ denote the TV ball of radius $\varepsilon$ around $P_{\text{int}}^0$ in the marginal $W$-space. As the PFN (parametrized by $\theta^t$ in the round $t$ of fine-tuning) is micro-tuned at step $t$ via empirical samples by sampling $= \{X^{(i)}\}_{i=1}^{n_f}$ from $P_{\text{int}}^0$ and optimizing $q(W \mid \theta^t)$. Then, with probability at least $1 - \delta$ over the sampling of $X^{(t)}$, the local gereralization capability of tuned PFN holds for any interventional distribution $P$ with distance from $P_{\text{int}}^0$ is exhibited in below:*

$$\sup_{P \in \mathcal{B}_{\varepsilon}(P_{\text{int}}^0)} \left|\hat{Q}_t - Q^W(P)\right| \leq \underbrace{\Delta_{\text{opt}}^t}_{\text{optimization bias}} + \underbrace{M_q \sqrt{\frac{2\log(2/\delta)}{n_f}}}_{\text{sampling error}} + \underbrace{2M_q \varepsilon}_{\text{TV-ball drift}}, \tag{28}$$

*where $\Delta_{\text{opt}}^t := |Q^W(P_{\text{int}}^0) - Q^W(q(W \mid \theta^t))|$ is the optimization bias.*

*Proof.* We decompose the total error via the triangle inequality:

$$\left|\hat{Q}_t - Q^W(P)\right| \leq \underbrace{\left|\hat{Q}_t - Q^W(P_{\text{int}}^0)\right|}_{\text{empirical / optimization error}} + \underbrace{\left|Q^W(P_{\text{int}}^0) - Q^W(P)\right|}_{\text{TV-ball deviation}}, \quad \forall P \in \mathcal{B}_{\varepsilon}(P_{\text{int}}^0).$$

**Step 1: TV-ball deviation.** For any $P \in \mathcal{B}_{\varepsilon}(P_{\text{int}}^0)$, by the standard bound of total variation distance for bounded functions:

$$\left|Q^W(P_{\text{int}}^0) - Q^W(P)\right| = \left|\int q(w)\,(P_{\text{int}}^{0,W} - P_W)(dw)\right| \leq 2M_q\,\text{TV}(P_{\text{int}}^{0,W}, P_W) \leq 2M_q\,\varepsilon.$$

**Step 2: Empirical / optimization error.** By definition, $\hat{Q}_t$ is an empirical average over $n_f$ i.i.d. samples from $P_{\text{int}}^0$:

$$\hat{Q}_t = \frac{1}{n_f} \sum_{i=1}^{n_f} q(W^{(i)}; \theta^t),$$

with $\mathbb{E}[\hat{Q}_t] = Q^W(q(W \mid \theta^t))$. We can bound the deviation of the empirical mean from its expectation via Hoeffding inequality:

$$\left|\frac{1}{n_f} \sum_{i=1}^{n_f} q(W^{(i)}; \theta^t) - Q^W(q(W \mid \theta^t))\right| \leq M_q \sqrt{\frac{2\log(2/\delta)}{n_f}}, \quad \text{w.p. } 1 - \delta.$$

Adding the optimization bias

$$\Delta_{\text{opt}}^t := |Q^W(P_{\text{int}}^0) - Q^W(q(W \mid \theta^t))|$$

yields

$$\left|\widehat{Q}_t - Q^W(P_{\text{int}}^0)\right| \le \Delta_{\text{opt}}^t + M_q \sqrt{\frac{2\log(2/\delta)}{n_f}}.$$

**Step 3: Combine bounds.** Combining Step 1 and Step 2, we have for all $P \in \mathcal{B}_\varepsilon(P_{\text{int}}^0)$:

$$\left|\widehat{Q}_t - Q^W(P)\right| \le \underbrace{\Delta_{\text{opt}}^t}_{\text{optimization bias}} + \underbrace{M_q \sqrt{\frac{2\log(2/\delta)}{n_f}}}_{\text{sampling error}} + \underbrace{2M_q\varepsilon}_{\text{TV-ball drift}}.$$

Taking the supremum over the TV ball yields the stated bound (28). □

**Lemma 2.** (*$k$-Center Approximation Guarantee*) *Let $\mathcal{S}_{\text{greedy}}$ be the set of $K$ interventional distributions selected by the greedy $k$-center algorithm under the distance $d(P, P') = \text{TV}(P_W, P'_W)$. Then the induced coverage radius satisfies $\varepsilon(\mathcal{S}_{\text{greedy}}) \le 2\varepsilon(\mathcal{S}^\star)$, where $\mathcal{S}^\star$ denotes an optimal solution to (15).*

*Proof.* Let $\mathcal{P}$ denote the candidate set of interventional distributions, and define the coverage radius induced by a selected set $\mathcal{S}$ as

$$\varepsilon(\mathcal{S}) = \sup_{P \in \mathcal{P}} \min_{Q \in \mathcal{S}} d(P, Q). \tag{29}$$

Since $d(P, P') = \text{TV}(P_W, P'_W)$ is a metric, it satisfies the triangle inequality.

Let $\mathcal{S}_{\text{greedy}} = \{P^{(1)}, \dots, P^{(K)}\}$ be the set selected by the greedy $k$-center algorithm, and denote $r = \varepsilon(\mathcal{S}_{\text{greedy}})$. Consider the next point that would be selected by the greedy rule:

$$P^{(K+1)} \in \arg\max_{P \in \mathcal{P}} \min_{Q \in \mathcal{S}_{\text{greedy}}} d(P, Q). \tag{30}$$

By definition, its distance to the nearest selected center is exactly $r$, namely $\min_{1 \le \ell \le K} d(P^{(K+1)}, P^{(\ell)}) = r$. Moreover, for any $1 \le i < j \le K + 1$, when $P^{(j)}$ was selected, its distance to the previously selected centers was the maximum remaining distance at that step. Since the coverage radius is non-increasing as more centers are added, we have

$$d(P^{(i)}, P^{(j)}) \ge r. \tag{31}$$

Therefore, the $K + 1$ points $\{P^{(1)}, \dots, P^{(K)}, P^{(K+1)}\}$ are pairwise separated by at least $r$. Now let $\mathcal{S}^\star$ be an optimal solution of size $K$. Assign each of the above $K + 1$ points to its nearest center in $\mathcal{S}^\star$. Since there are $K + 1$ points but only $K$ optimal centers, by the pigeonhole principle, there must exist two distinct points $P^{(i)}$ and $P^{(j)}$ assigned to the same optimal center $P^\star \in \mathcal{S}^\star$. By the triangle inequality,

$$d(P^{(i)}, P^{(j)}) \le d(P^{(i)}, P^\star) + d(P^\star, P^{(j)}) \le \varepsilon(\mathcal{S}^\star) + \varepsilon(\mathcal{S}^\star) = 2\varepsilon(\mathcal{S}^\star).$$

On the other hand, the pairwise separation property gives

$$r \le d(P^{(i)}, P^{(j)}). \tag{32}$$

Combining the above two inequalities yields

$$r \le 2\varepsilon(\mathcal{S}^\star). \tag{33}$$

Since $r = \varepsilon(\mathcal{S}_{\text{greedy}})$, we obtain

$$\varepsilon(\mathcal{S}_{\text{greedy}}) \le 2\varepsilon(\mathcal{S}^\star). \tag{34}$$

This completes the proof. □

**Theorem 6.** (**Local Generalization under MSF**) *Let $\mathcal{S}_{\text{greedy}} \subseteq \mathcal{S}_{\text{int}}$ be a set of $K$ interventional distributions selected by the greedy $k$-center algorithm under the distance $d(P, P') = \text{TV}(P_W, P'_W)$, and let $\varepsilon(\mathcal{S}_{\text{greedy}})$ denote its induced coverage*

*radius. Suppose that the PFN is fine-tuned using $n_f$ samples drawn from the mixture distribution supported on $\mathcal{S}_{\text{greedy}}$. Then, with probability at least $1 - \delta$, for any interventional distribution $P \in \mathcal{S}_{\text{int}}$, the following bound holds:*

$$\left|\widehat{Q}_t - Q^W(P)\right| \leq \Delta^t_{\text{opt}} + M_q\sqrt{\frac{2\log(2/\delta)}{n_f}} + 4M_q\,\varepsilon(\mathcal{S}^\star), \tag{35}$$

*where $\mathcal{S}^\star$ denotes an optimal solution to the $k$-center objective in (15).*

*Proof.* Fix any interventional distribution $P \in \mathcal{S}_{\text{int}}$. By definition of the coverage radius $\varepsilon(\mathcal{S}_{\text{greedy}})$, there exists a selected distribution $P' \in \mathcal{S}_{\text{greedy}}$ such that

$$\text{TV}(P_W, P'_W) \leq \varepsilon(\mathcal{S}_{\text{greedy}}).$$

Consider the PFN fine-tuned on samples drawn from the mixture distribution supported on $\mathcal{S}_{\text{greedy}}$. Since $P'$ belongs to the support of the fine-tuning distribution, the point-wise local generalization guarantee in Theorem 5 applies to $P$ relative to $P'$. Specifically, with probability at least $1 - \delta$, we have

$$\left|\widehat{Q}_t - Q^W(P)\right| \leq \Delta^t_{\text{opt}} + M_q\sqrt{\frac{2\log(2/\delta)}{n_f}} + 2M_q\,\text{TV}(P_W, P'_W).$$

Substituting the bound on $\text{TV}(P_W, P'_W)$ yields

$$\left|\widehat{Q}_t - Q^W(P)\right| \leq \Delta^t_{\text{opt}} + M_q\sqrt{\frac{2\log(2/\delta)}{n_f}} + 2M_q\,\varepsilon(\mathcal{S}_{\text{greedy}}).$$

Finally, by the $k$-center approximation guarantee in Lemma 2, the greedy selection satisfies $\varepsilon(\mathcal{S}_{\text{greedy}}) \leq 2\varepsilon(\mathcal{S}^\star)$. Combining the above inequalities completes the proof. $\square$

## B.2. Examples of Local Generalizations of PWF

**Example 2** (Local Generalization on Linear–Gaussian Models)**.** *Consider a linear-Gaussian causal model on variables $X$ with structural matrix $B$ and zero-mean Gaussian exogenous noises: $X = BX + \varepsilon$ with $\varepsilon \sim \mathcal{N}(0, D)$, where $D = \text{diag}(\sigma_1^2, \ldots, \sigma_n^2)$. Then the $W$-marginal distributions admits Gaussian, i.e., $P_{\text{int}}^{W|\text{Do}(S=s)} = \mathcal{N}(\mu_S, \Sigma_S)$ and $P_{\text{int}}^{W|\text{Do}(T=t)} = \mathcal{N}(\mu_T, \Sigma_T)$. Let $M$ be the linear map from intervention values to target means (so $\mu_S - \mu_T = M(s - t)$).*

$$\boxed{\text{TV}_W\left(P_{\text{int}}^{W|\text{Do}(S=s)}, P_{\text{int}}^{W|\text{Do}(T=t)}\right) \leq \sqrt{\tfrac{1}{2}\,\text{KL}\big(\mathcal{N}(\mu_S, \Sigma_S)\,\|\,\mathcal{N}(\mu_T, \Sigma_T)\big)},}$$

*where the Gaussian KL on the right is the standard closed form (see proof).*

*Detailed proof.* We prove three facts in order and then combine them to obtain the TV bound.

**Notation and setup.** Suppose that $(I - B)$ invertible. Let $S \subseteq \{1, \ldots, M\}$ and $T \subseteq \{1, \ldots, M\}$ be intervention index sets. Fix a target index set $W \subseteq \{1, \ldots, M\}$ such that both interventional marginals on $W$ are well-defined; for concreteness assume $W \subseteq -S \cap -T$ (i.e. $W$ are not directly intervened on). Moreover, we write $-S$ for the complement of $S$, and use similar notation for $-T$. We let $R_{W,-S}$ denote the selection/projection matrix that extracts the $W$-coordinates from $X_{-S}$.

**Step 1 — Mean and covariance under a hard intervention.** Replace structural equations for indices in $S$ by constants $s$. Partition variables as $X = (X_{-S}, X_S)$. The subsystem for the non-intervened block $X_{-S}$ is

$$X_{-S} = B_{-S,-S}\,X_{-S} + B_{-S,S}\,s + \varepsilon_{-S}.$$

Solve for $X_{-S}$:

$$X_{-S} = (I - B_{-S,-S})^{-1}B_{-S,S}\,s + (I - B_{-S,-S})^{-1}\varepsilon_{-S}.$$

Projecting to coordinates $W \subseteq -S$ via $R_{W,-S}$ yields the $W$-marginal under $\mathrm{Do}(S=s)$:

$$\mu_S = R_{W,-S}(I - B_{-S,-S})^{-1}B_{-S,S}\, s, \qquad \Sigma_S = R_{W,-S}(I - B_{-S,-S})^{-1}D_{-S,-S}\big((I - B_{-S,-S})^{-1}\big)^{\top} R_{W,-S}^{\top}.$$

(Here $D_{-S,-S}$ is the principal submatrix of $D$ for indices $-S$.) Analogous formulas hold for $\mu_T, \Sigma_T$ by replacing $S, t$ with $T, t$ and using blocks indexed by $-T$.

**Step 2 — Linear relation for the mean difference.** Assuming $W \subseteq -S \cap -T$ so both $\mu_S, \mu_T$ are defined by the above form with projections from the same coordinate set, we can write their difference as

$$\mu_S - \mu_T = \Big[R_{W,-S}(I - B_{-S,-S})^{-1}B_{-S,S}\Big]s - \Big[R_{W,-T}(I - B_{-T,-T})^{-1}B_{-T,T}\Big]t.$$

In many standard orderings (or when $W, -S, -T$ coincide for the projection step) this reduces to the compact representation

$$\mu_S - \mu_T = M(s - t),$$

with $M$ the appropriate linear map from intervention coordinates to $W$-means; for instance, when the blocks align under the partition $X = (W, S, R)$, one may take $M = (I - B_{WW})^{-1}B_{W,S}$, recovering the often-used expression $\mu_S - \mu_T = M(s - t)$. (If $-S$ and $-T$ differ, one must embed coordinates appropriately; the relation remains linear in $s$ and $t$.)

**Step 3 — TV bound via Pinsker and Gaussian KL (no equal-covariance assumption).** Both target marginals are multivariate Gaussian:

$$P_{\mathrm{int}}^{W|\mathrm{Do}(S=s)} = \mathcal{N}(\mu_S, \Sigma_S), \qquad P_{\mathrm{int}}^{W|\mathrm{Do}(T=t)} = \mathcal{N}(\mu_T, \Sigma_T).$$

For any two probability measures $P, Q$ we have Pinsker's inequality

$$\mathrm{TV}(P,Q) \leq \sqrt{\tfrac{1}{2}\mathrm{KL}(P\|Q)}.$$

Applying with $P = \mathcal{N}(\mu_S, \Sigma_S), Q = \mathcal{N}(\mu_T, \Sigma_T)$ yields

$$\mathrm{TV}_W\big(P_{\mathrm{int}}^{W|\mathrm{Do}(S=s)},\, P_{\mathrm{int}}^{W|\mathrm{Do}(T=t)}\big) \leq \sqrt{\tfrac{1}{2}\,\mathrm{KL}\big(\mathcal{N}(\mu_S, \Sigma_S)\,\|\,\mathcal{N}(\mu_T, \Sigma_T)\big)}.$$

The KL divergence between multivariate Gaussians is (standard):

$$\mathrm{KL}\big(\mathcal{N}(\mu_S, \Sigma_S)\,\|\,\mathcal{N}(\mu_T, \Sigma_T)\big) = \tfrac{1}{2}\Big\{ \mathrm{tr}(\Sigma_T^{-1}\Sigma_S) - d + \ln\frac{\det \Sigma_T}{\det \Sigma_S} \quad + (\mu_T - \mu_S)^{\top}\Sigma_T^{-1}(\mu_T - \mu_S)\Big\},$$

where $d = \dim(W)$. This expression depends explicitly on $(\mu_S, \Sigma_S), (\mu_T, \Sigma_T)$, hence (via the formulas in Step 1) is computable from the SEM matrices $B, D$ and the intervention values $s, t$.

Combining the last two displays gives the boxed TV upper bound stated in the Example.

$$\square$$

**Example 3** (Local Generalization on Additive Non-linear Models (ANM)). *Let the SCM follow an ANM: each structural equation takes the form $V_i = f_i(\mathrm{Pa}(V_i)) + \varepsilon_i$ with mutually independent noise variables $\varepsilon_i$. For each intervened variable $S_j$, define the maximal functional range $R_{S,j} := \sup_{u,u' \in \mathcal{U}_S}|f_{S_j}(\mathrm{Pa}(S_j; u)) - f_{S_j}(\mathrm{Pa}(S_j; u'))|$, and analogously $R_{T,\ell}$ for $T_\ell$.*

(a) *(**Sub-Gaussian exogenous noises**). Assume that the exogenous noises for variables in $S$ and $T$ are sub-Gaussian with parameters $\sigma_{S,j}$ and $\sigma_{T,\ell}$, respectively. Define*

$$\Delta_{S,j}^{(G)} := \frac{1}{\sqrt{2\pi}\sigma_{S,j}}\Big(1 - \exp\Big(-\frac{R_{S,j}^2}{2\sigma_{S,j}^2}\Big)\Big), \qquad \Delta_{T,\ell}^{(G)} := \frac{1}{\sqrt{2\pi}\sigma_{T,\ell}}\Big(1 - \exp\Big(-\frac{R_{T,\ell}^2}{2\sigma_{T,\ell}^2}\Big)\Big).$$

*Let $\tfrac{1}{2}\sum_j \Delta_{S,j}^{(G)} = \varepsilon_s$ and $\tfrac{1}{2}\sum_\ell \Delta_{T,\ell}^{(G)} = \varepsilon_t$, then*

$$\mathrm{TV}_W\big(P_{\mathrm{int}}^{W|\mathrm{Do}(S=s)}, P_{\mathrm{int}}^{W|\mathrm{Do}(T=t)}\big) \leq \varepsilon_s + \varepsilon_t.$$

*(b)* ***(Bounded-support & Lipschitz densities).*** *Suppose each relevant noise density $p_\varepsilon$ is L-Lipschitz on its support. Define $\Delta_{S,j}^{(B)} := 2L_{S,j}R_{S,j}$ and $\Delta_{T,\ell}^{(B)} := 2L_{T,\ell}R_{T,\ell}$. Let $\frac{1}{2}\sum_j \Delta_{S,j}^{(B)} = \varepsilon_s$ and $\frac{1}{2}\sum_\ell \Delta_{T,\ell}^{(B)} = \varepsilon_t$, then again*

$$\mathrm{TV}_W(P_{\mathrm{int}}^{W|\mathrm{Do}(S=s)}, P_{\mathrm{int}}^{W|\mathrm{Do}(T=t)}) \leq \varepsilon_s + \varepsilon_t.$$

*Proof.* The divergence between interventional outcomes on $W$ is entirely governed by the functional range and noise smoothness of the upstream intervention nodes $S, T$. The result quantifies how local changes in the mechanisms of $S, T$ propagate through the ANM to shift the downstream distribution of $W$, and thereby control any causal query $Q(P_{\mathrm{int}}) = \int q(w)\, P_{\mathrm{int},W}(dw)$ through the bound $|Q(P^{(S)}) - Q(P^{(T)})| \leq 2M_q\, \mathrm{TV}_W(P^{(S)}, P^{(T)})$. We establish the bound in three steps.

**Step 1. Reduction via the triangle inequality.** For the three distributions $P_W^{(S)} := P_{W|\mathrm{Do}(S=s)}$, $P_W^{(T)} := P_{W|\mathrm{Do}(T=t)}$, and $P_W := P_{W,\mathrm{obs}}$, the triangle inequality for total variation gives

$$\mathrm{TV}_W(P_W^{(S)}, P_W^{(T)}) \leq \mathrm{TV}_W(P_W^{(S)}, P_W) + \mathrm{TV}_W(P_W, P_W^{(T)}). \tag{T$'$}$$

Hence it suffices to bound each single-intervention term $\mathrm{TV}_W(P_{W|\mathrm{Do}(A=a)}, P_W)$ for a generic intervention set $A \subseteq X$.

**Step 2. Single-intervention bound (general $A$).** Fix $A \subseteq X$ and let $U_A = \bigcup_j \mathrm{Pa}(A_j)$ be the union of parents of variables in $A$. By the $g$-formula and ANM factorization,

$$p_{W|\mathrm{Do}(A=a)}(w) = \int_{u \in \mathcal{U}_A} p(w \mid A = a, U_A = u)\, p_{U_A}(u)\, du.$$

Similarly, the observational conditional reads

$$p_{W|A=a}(w) = \int_{u \in \mathcal{U}_A} p(w \mid A = a, U_A = u)\, p_{U_A|A=a}(u)\, du.$$

Therefore,

$$\begin{aligned}
\mathrm{TV}_W(p_{W|\mathrm{Do}(A=a)}, p_W) &\leq \frac{1}{2}\int_{w,u} p(w \mid a, u)\,\big|p_{U_A}(u) - p_{U_A|A=a}(u)\big|\, du\, dw \\
&= \frac{1}{2}\int_u \big|p_{U_A}(u) - p_{U_A|A=a}(u)\big|\, du = \mathrm{TV}(p_{U_A}, p_{U_A|A=a}),
\end{aligned} \tag{1$'$}$$

since $p(w \mid a, u)$ integrates to one. Thus, the divergence on the *target set* $W$ is upper bounded by how much the parental variables $U_A$ deviate under the intervention on $A$.

**Step 3. Bounding the parent-level shift.** From Bayes' rule and the ANM structure,

$$p_{U_A|A=a}(u) = \frac{\prod_j p_{\varepsilon_{A_j}}(a_j - f_{A_j}(\mathrm{Pa}(A_j; u)))\, p_{U_A}(u)}{p_A(a)}.$$

Following the product-difference argument as in the $Z$-level proof,

$$\mathrm{TV}(p_{U_A}, p_{U_A|A=a}) \leq \frac{1}{2}\sum_j \sup_{u,u' \in \mathcal{U}_A} \big|p_{\varepsilon_{A_j}}(a_j - f_{A_j}(\mathrm{Pa}(A_j; u))) - p_{\varepsilon_{A_j}}(a_j - f_{A_j}(\mathrm{Pa}(A_j; u')))\big|. \tag{2$'$}$$

We now apply two specific noise assumptions.

*(a) Sub-Gaussian case.* For Gaussian (or sub-Gaussian upper-bounded) noise with parameter $\sigma_{A_j}$, the density difference satisfies

$$|\varphi_\sigma(t) - \varphi_\sigma(t')| \leq \tfrac{1}{\sqrt{2\pi}\sigma}(1 - e^{-(t-t')^2/(2\sigma^2)}).$$

Let $R_{A,j}$ denote the maximal amplitude of $f_{A_j}$ on its domain; plugging into (2$'$) gives

$$\mathrm{TV}_W(p_{W|\mathrm{Do}(A=a)}, p_W) \leq \tfrac{1}{2}\sum_j \frac{1}{\sqrt{2\pi}\sigma_{A_j}}\left(1 - e^{-R_{A,j}^2/(2\sigma_{A_j}^2)}\right).$$

Setting $A = S$ and $A = T$ and combining via (T′) yields

$$\mathrm{TV}_W(P_{\mathrm{int}}^{W|\mathrm{Do}(S=s)}, P_{\mathrm{int}}^{W|\mathrm{Do}(T=t)}) \leq \varepsilon_s + \varepsilon_t.$$

*(b) Bounded-support & Lipschitz case.* If $p_{\varepsilon_{A_j}}$ is $L_{A_j}$-Lipschitz, then

$$|p_{\varepsilon_{A_j}}(a_j - f_{A_j}(u)) - p_{\varepsilon_{A_j}}(a_j - f_{A_j}(u'))| \leq L_{A_j}|f_{A_j}(u) - f_{A_j}(u')| \leq L_{A_j} R_{A,j}.$$

Substituting into (2′) and then (T′) yields

$$\mathrm{TV}_W(P_{\mathrm{int}}^{W|\mathrm{Do}(S=s)}, P_{\mathrm{int}}^{W|\mathrm{Do}(T=t)}) \leq \varepsilon_s + \varepsilon_t.$$

$\square$

**Corollary 9** (Local Generalization of Causal Query $Q$). *We can prove the local generalization property of the interventional-finetuned PFN model below:*

*(a)* *(**Sub-Gaussian noises**). If*

$$\tfrac{1}{2}\sum_j \Delta_{S,j}^{(G)} \leq \varepsilon_s, \qquad \tfrac{1}{2}\sum_\ell \Delta_{T,\ell}^{(G)} \leq \varepsilon_t,$$

*(with $\Delta^{(G)}$ defined as in the theorem) then*

$$|Q(P^S) - Q(P^T)| \leq 2\|q\|_\infty(\varepsilon_s + \varepsilon_t).$$

*(b)* *(**Bounded-support & Lipschitz densities**). If*

$$\tfrac{1}{2}\sum_j \Delta_{S,j}^{(B)} \leq \varepsilon_s, \qquad \tfrac{1}{2}\sum_\ell \Delta_{T,\ell}^{(B)} \leq \varepsilon_t,$$

*(with $\Delta^{(B)}$ defined as in the theorem) then*

$$|Q(P^S) - Q(P^T)| \leq 2\|q\|_\infty(\varepsilon_s + \varepsilon_t).$$

*Proof.* Based on proof of Theorem 5, one gets explicit $\varepsilon_s, \varepsilon_t$ satisfying $\mathrm{TV}_W(P_{(S=s)}, P_{(T=t)}) \leq \varepsilon_s + \varepsilon_t$. Finally, since $Q$ is linear with $\|q\|_\infty = M_q < \infty$,

$$|Q^W(P_{\mathrm{int}}^S) - Q^W(P_{\mathrm{int}}^T)| = \left| \int q(w)\, [p_{\mathrm{int}}^{W|\mathrm{Do}(S=s)} - p_{\mathrm{int}}^{W|\mathrm{Do}(T=t)}](dw) \right| \leq 2M_q\, \mathrm{TV}\Big(P_{\mathrm{int}}^{W|\mathrm{Do}(S=s)}, P_{\mathrm{int}}^{W|\mathrm{Do}(T=t)}\Big). \quad (36)$$

and substituting the target-level TV bound yields the stated inequality. $\square$

## C. Experimental Details

### C.1. Setup

We evaluate three PFN-based tabular backbones (i.e., TabPFN v2.5 (Hollmann et al.), CausalPFN (Balazadeh et al., 2025), and DoPFN (Robertson et al., 2025)) and adapt each of them via the same fine-tuning protocols used in our experiments (Pointwise fine-tuning and Meta-Sample fine-tuning; see Sec. 5). Both fine-tuning experiments are implemented in PyTorch and conducted on two NVIDIA H100 GPUs. We finetune each PFN backbone using standard gradient-based optimization on interventional regression tasks generated from three SCM settings (i.e., linear, nonlinear, and interaction) with additive noise $\varepsilon_Y \sim \mathcal{N}(0, 0.1^2)$. During training, the batch size per gradient step is $B = 32$.

We construct a pool of interventional distributions $\mathcal{P} \subset \mathcal{S}_{\mathrm{int}} = [30, 60, 90, 100]$. Data are generated in a task-based manner. Each task consists of a context set of 128 samples and a query set of 32 samples. As a result, each interventional distribution yields 160 samples per task, and 8,000 samples in total when 50 tasks are sampled. Each interventional distribution corresponds to a do-configuration that performs *hard interventions* $\mathrm{do}(X_i = a, X_j = b)$ on two randomly chosen variables

$X_i$ and $X_j$, with intervention values $(a, b)$ independently sampled from $\{-3, -2, -1, 1, 2, 3\}$. To mitigate the impact of randomness, we repeat each experiment 10 times with different random seeds and report the average results.

Throughout all SCMs, the feature vector $X \in \mathbb{R}^d$ is sampled from a multivariate Gaussian distribution, $X \sim \mathcal{N}(0, \Sigma)$. To introduce structured correlations between features, we instantiate the covariance matrix $\Sigma$ with a Toeplitz structure: $\Sigma_{ij} = \rho^{|i-j|}$, where we set $\rho = [0.5, 0.7, 0.9]$. This yields a valid covariance matrix with decaying correlations as the distance between feature indices increases. The outcome $Y$ is then generated via the following SCM equations of increasing complexity.

- **Linear SCM:** The outcome is a linear combination of features

$$Y = \mathbf{X}^\top \beta + \varepsilon_Y, \quad \varepsilon_Y \sim \mathcal{N}(0, 0.1^2). \tag{37}$$

- **Non-linear Additive SCM:** To test the model's ability to handle non-linearity without complex feature dependencies, we define the outcome as:

$$Y = X_0 X_1 + \tanh\Big(\sum_{k=2}^{i} X_k\Big) + \sin\Big(\sum_{k=i+2}^{i+1+j} X_k\Big) + \varepsilon_Y, \tag{38}$$

  where $\varepsilon_Y \sim \mathcal{N}(0, 0.1^2)$ and $(i, j)$ are the corresponding indices for the chosen dimension of variable $m$ (e.g., $i = \lfloor m/2 \rfloor$, $j = m - i - 2$). This model incorporates heterogeneous non-linear effects over multiple feature subsets while avoiding dense cross-dimensional interactions.

- **Interaction SCM:** To simulate complex, high-dimensional dependencies across interventions, we consider the SCM with strong pairwise interactions:

$$Y = \alpha^\top X + \sum_{i<j} \gamma_{ij} X_i X_j + \varepsilon_Y, \; \varepsilon_Y \sim \mathcal{N}(0, 0.1^2). \tag{39}$$

For real-world datasets, we leveraged the RealCause framework (Neal et al., 2020) based on the Lalonde study to construct a semi-synthetic data generation pipeline. We first consolidated the original covariates and treatment assignments into a unified set of intervenable candidate variables. To model complex nonlinear causal mechanisms, a randomly initialized neural network was employed as the outcome generator. We implemented a hierarchical stochastic intervention strategy: specifically, we randomly selected a subset of samples with probability $\alpha = [0.5, 0.8, 1.0]$ and subsequently perturbed a proportion $\beta = [02, 0.3, 0.4]$ of feature dimensions within these samples. The perturbed features $x$ were substituted via uniform resampling from their respective empirical ranges $[x_{min}, x_{max}]$. Ultimately, these modified inputs were mapped through the generator to synthesize the final observed outcomes $Y$.

**Parameter Configurations.** For all three types of SCMs, we vary the scale of SCM $M \in [4, 6, 8, 10]$. During the DGP process, data are generated in a task-based manner. For each interventional distribution, 50 tasks are sampled, each task with 160 query samples, resulting in 8,000 samples per distribution. The size of the interventional distribution set $\mathcal{S}_{\text{int}}$ scales with $M$ as $[30, 60, 90, 100]$. To ensure that compared PFN baselines are pretrained on a diversity of interventional/observational distributions, we further select a subset of $\mathcal{S}_{\text{int}}$ to finetune these base models, serving as pre-trained PFNs in the regime of general interventions. In concrete, $8,000$ samples are sampled from each pre-trained interventional distribution, and $[15, 30, 50, 60]$ interventional distributions are selected for pre-training.

### C.2. Implementation of Pointwise fine-tuning

The pointwise fine-tuning procedure consists of two stages:

- Pretraining a PFN backbone on a subset of interventional distributions.
- fine-tuning the pretrained PFN to an uncovered interventional distribution.

**Stage 1 (Pretraining).** We select $K_{\text{pre}} \in [15, 30, 50, 60]$ interventional distributions from the intervention pool $\mathcal{P}$, where $|\mathcal{P}| \in [30, 60, 90, 100]$ depends on the SCM scale. These distributions are induced by distinct do-configurations under the same SCM. The PFN backbone is trained for 10 epochs using AdamW with learning rate $2 \times 10^{-5}$ and weight decay $2 \times 10^{-4}$, resulting in a pretrained PFN that has not observed the remaining interventional distributions.

**Stage 2 (Pointwise Adaptation).** From the set of interventional distributions not used in Stage 1, we select a target interventional distribution and denote it as $D_1$. We sample 50 tasks exclusively from $D_1$ and further finetune the pretrained

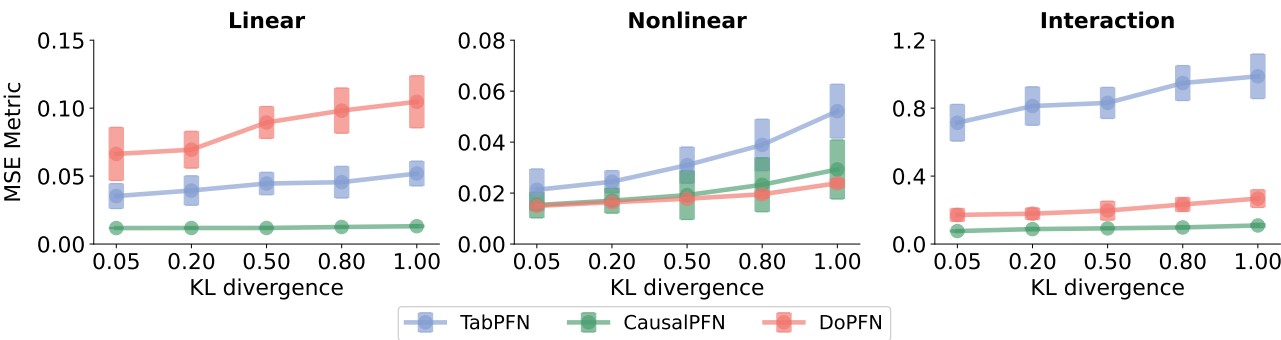

*Figure 8.* Local Generalization of our PWF strategy on the Lalonde Dataset for each baseline, where x-axis refers to the testing interventional distributions with Top-K small divergence (KL) from the fine-tuned distribution.

PFN for 10 epochs, using AdamW with learning rate $2 \times 10^{-5}$ and zero weight decay to enable stronger local adaptation. Gradient clipping with maximum norm 1.0 is applied in both stages.

**Evaluation.** For evaluation, we additionally consider two other uncovered interventional distributions. Among them, $D_2$ denotes another interventional distribution that is similar to $D_1$, while $D_3$ represents an interventional distribution that is different from $D_1$. The similarity between interventional distributions is quantified by estimating the KL divergence between the *marginal distributions of the outcome $Y$* generated under their respective do-configurations (using samples from the data pool). All distributions $D_1, D_2$, and $D_3$ are induced by do-interventions under the same SCM. The specific parameterization and the resulting distributions are as follows:

- **Linear SCM:** The coefficient vector $\beta \in \mathbb{R}^d$ is sampled from a Gaussian distribution, $\beta \sim \mathcal{N}(0, I_m)$. Under a hard intervention $\mathrm{do}(X_i = a, X_j = b)$, the interventional mean of $Y$ is $\mu = a\beta_i + b\beta_j$, and its variance is $\beta^\top \Sigma \beta + \epsilon_Y$. The intervention values $(a, b)$ are sampled from $\{-3, -2, -1, 1, 2, 3\}$, leading to $\mu$ varying approximately within $(-2, 2)$ given the typical scale of $\beta$. Thus, the outcome follows $Y \sim \mathcal{N}(\mu, \beta^\top \Sigma \beta + \epsilon_Y)$.
- **Non-linear Additive SCM:** $Y = X_0 X_1 + \tanh(\sum_{k=2}^{i} X_k) + \sin(\sum_{k=i+1}^{i+j} X_k) + \varepsilon_Y$. Here, the indices partition the features: $i = \lfloor m/2 \rfloor$ and $j = m - i - 2$, creating three functional groups. Under a hard intervention $\mathrm{do}(X_i = a, X_j = b)$, we sample the unaffected variables $X_{\neg\{i,j\}}$ from their conditional Gaussian distribution given the intervened values, then compute $Y$ via the structural equation. The resulting distribution of $Y$ is determined by the product term $X_0 X_1$, the saturated $\tanh$ transform, and the saturated $\sin$ transform applied to the conditionally Gaussian inputs.
- **Interaction SCM:** $Y = \alpha^\top X + \sum_{i<j} \gamma_{ij} X_i X_j + \varepsilon_Y$. We set the linear coefficients $\alpha$ to $[0.5, 1, 1.5, 2]$ (cyclically repeated if $m > 4$). The interaction coefficients $\gamma_{ij}$ are sampled such that only **30%** of all possible pairwise interactions are non-zero; each non-zero $\gamma_{ij}$ is drawn independently from $\mathcal{N}(0, 0.5^2)$. Under a hard intervention $\mathrm{do}(X_i = a, X_j = b)$, we conditionally sample the unaffected variables $X_{\neg\{i,j\}}$ from their Gaussian distribution and compute $Y$.

### C.3. Implementation of Meta-Sample fine-tuning

Meta-sample fine-tuning is performed on a pretrained PFN backbone. We use the same set of interventional distributions $\mathcal{P}$ and data generation process as described in the Pointwise fine-tuning setup (Sec. C). For each interventional distribution $P \in \mathcal{P}$, we compute a $2D$ moment embedding based on the mean and standard deviation of predicted query outputs. We then apply $k$-greedy selection over the embedding space to select $K$ interventional distributions, where $K \in [5, 7, 9, 11]$. For each selected distribution, we sample 50 tasks per epoch, and train for 10 epochs in total. fine-tuning is conducted using AdamW with learning rate $2 \times 10^{-5}$, zero weight decay, and gradient clipping with maximum norm 1.0.

### C.4. Experiments on Lalonde$_{\text{PSID}}$ Dataset

The Lalonde$_{\text{PSID}}$ dataset consists of 100 heterogeneous tables, each corresponding to a distinct data-generating process with interventions. We conduct experiments using both pointwise fine-tuning (PWF) and meta-sample fine-tuning (MSF) to evaluate model robustness under distributional variation across tables.

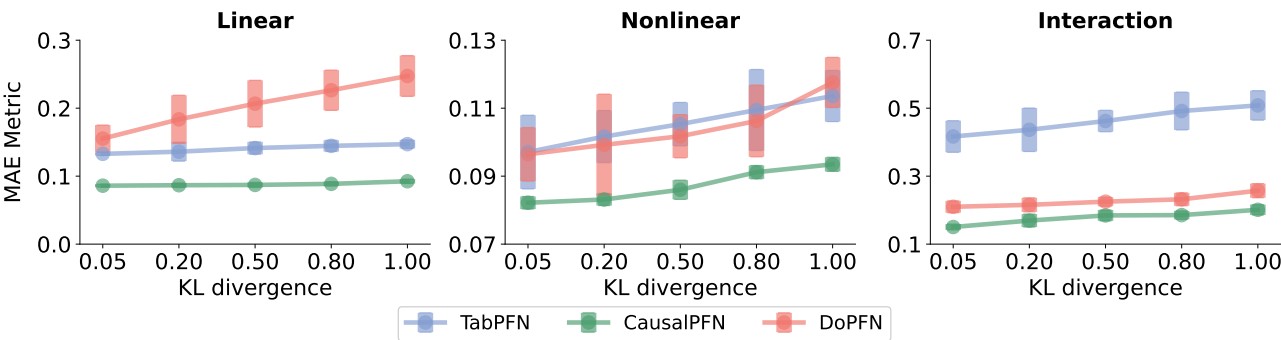

*Figure 9.* Local Generalization of our PWF strategy on the Lalonde Dataset for each baseline, where x-axis refers to the testing interventional distributions with Top-K small divergence (KL) from the fine-tuned distribution.

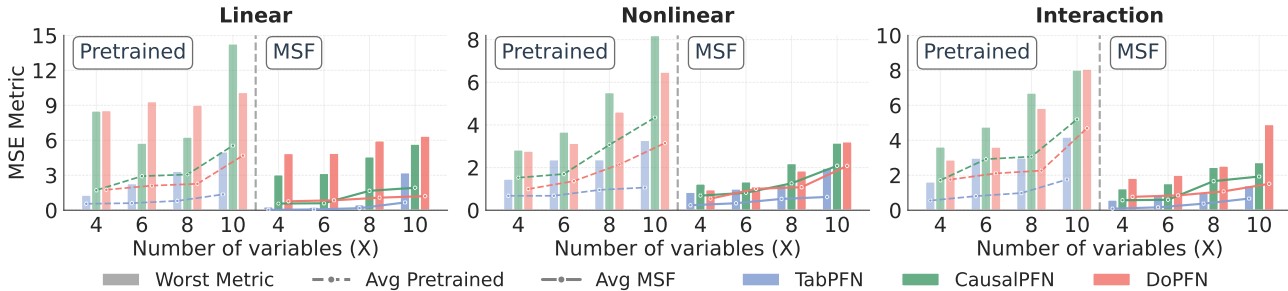

*Figure 10.* Uniform generalization assessment of MSF across different variable scales. The figure compares the mean and average counterfactual prediction results (MSE) of pretrained zero-shot models against the MSF strategy over the entire set of uncovered interventional distributions in $\mathcal{S}_{\text{int}}$.

**PWF on Lalonde$_{\text{PSID}}$.** For PWF, we randomly select a single table from the 100 available tables and finetune the pretrained PFN model exclusively on this table. To systematically evaluate how performance generalizes across distributional shifts, we measure the discrepancy between tables using the Wasserstein-2 distance (Deshpande et al., 2019). Since exact computation of $W_2$ in high dimensions is computationally prohibitive, we adopt the Sliced Wasserstein-2 distance as a scalable approximation:

$$SW_2(P_i, P_j) = \left( \mathbb{E}_{v \sim \text{Unif}(\mathbb{S}^{d-1})} \left[ W_2^2(\langle P_i, v \rangle, \ \langle P_j, v \rangle) \right] \right)^{1/2}, \tag{40}$$

where $\langle P, v \rangle$ denotes the one-dimensional distribution obtained by projecting samples from $P$ onto direction $v$. In practice, the expectation is approximated via Monte-Carlo sampling over random projection directions.

Specifically, all remaining tables are ranked according to their Wasserstein-2 distance to the finetuned table, from nearest to farthest, and the performance of the finetuned PFN is evaluated along this ordering. fine-tuning is performed for 40 epochs using the AdamW optimizer with a learning rate of $1 \times 10^{-5}$.

**MSF on Lalonde$_{\text{PSID}}$.** For MSF, we apply a greedy $k$-center selection strategy over the 100 Lalonde$_{\text{PSID}}$ tables, using the Wasserstein-2 distance as the underlying metric. This procedure selects a diverse subset of tables that maximally covers the space of data-generating distributions. We consider subset sizes $K \in \{30, 35, 40, 45\}$. For each selected table, the model is finetuned for 20 epochs. All MSF experiments are conducted using the AdamW optimizer with a learning rate of $1 \times 10^{-5}$.

**Additional Experimental Results for Local Generalization.** Moreover, we leave detailed illustration of the local generalization property on our synthetic dataset in Figure 8 and 9. The similar trends are exhibited as in our main paper, further verifying the correctness of our Theorem on the local generalization property of PWF.

**Additional Experimental Results measured by the MSE Metric.** Finally, we leave the assessment under the MSE metric in the Figure 10, reporting similar trends of the uniform generalization of MSF.

*Table 4.* Performance on RecSim under multi-variable joint interventions. We report worst-case MAE with $K = 7, 7, 7, 35$.

| Dataset | Pre-trained | PWF | MSF |
|---|---|---|---|
| RecSim (KnownTarget, TabPFN) | 1.42 | 0.36 | 0.34 |
| RecSim (KnownTarget, DoPFN) | 1.26 | 0.30 | 0.29 |
| RecSim (KnownTarget, CausalPFN) | 1.19 | 0.25 | 0.24 |
| RecSim (UnknownTarget, TabPFN) | 1.42 | 1.12 | 0.22 |
| RecSim (UnknownTarget, DoPFN) | 1.26 | 1.00 | 0.17 |
| RecSim (UnknownTarget, CausalPFN) | 1.19 | 0.91 | 0.14 |

## D. Extension on the RecSim Dataset

**Real-world Data Generation.** We use the RecSim document-recommendation simulator. Each user is represented by a topic-affinity vector $X \in \mathbb{R}^d$, and each document has a topic $c_i$ and a quality score $q_i$. To induce confounding, we generate an intention vector as

$$L = X + \epsilon_L, \qquad \epsilon_L \sim \mathcal{N}(0, 0.81I). \tag{41}$$

Then each document is scored by $\text{Score}_i = L_{c_i} + q_i$, and the top-$s$ documents are selected to form the bundle treatment. Given the user features and the recommended bundle, the simulator outputs the ground-truth click rate. We evaluate performance by RMSE on an unbiased test set. The setup fixes $n = 10000$, $d = 4$, and $s = 4$, where user affinities and document qualities are sampled from standard normal distributions (Zou et al., 2020).

**Multi-variable Interventional Table Generation.** For each interventional table, we first sample a subset of variables $S_t \subseteq \{1, \ldots, d\}$, where $S_t$ can contain one or multiple variables and may vary across different tables. We then apply a hard intervention $do(X_{S_t} = x_{S_t}^{(t)})$ by fixing all variables in $S_t$ to prescribed values, while leaving the remaining variables unchanged. Given the intervened user feature vector, we generate the intention vector as

$$L = X + \epsilon_L, \qquad \epsilon_L \sim \mathcal{N}(0, 0.8I). \tag{42}$$

Each document is then scored by

$$\text{Score}_i = L_{c_i} + q_i, \tag{43}$$

and the top-$s$ scored documents form the bundle treatment. The simulator then outputs the corresponding ground-truth click rate. Each distinct intervention $do(X_{S_t} = x_{S_t}^{(t)})$ defines one interventional distribution and therefore one interventional table. Repeating this process with different intervention subsets and intervention values yields 100 interventional tables in total, denoted by $\{D^{(t)}\}_{t=1}^{100}$, where each table corresponds to a different interventional regime.

We conduct additional experiments in a real-world recommendation scenario and report the results in Table 4. Bundle recommendation serves as a natural case for multi-variable joint interventions, since platforms often modify multiple ranking controls jointly, such as several topic-affinity dimensions and bundle-ranking rules. Each joint policy defines a new interventional table, and PFN pre-training may fail once such tables are uncovered or out-of-distribution.

We use RecSim to simulate recommendation data. User topic affinities are treated as confounders, top-scored document bundles are regarded as treatments, and click-rate prediction is evaluated by RMSE on an unbiased test set. The results show that our fine-tuning approaches consistently improve pre-trained PFNs. In particular, MSF achieves strong performance under both known-target and unknown-target settings, demonstrating its robustness to uncovered joint interventions.

