# OpenReview forum: "Unveiling Prior-Data Fitted Networks on Causal Effect Estimation: Pre-Training or Fine-Tuning?"
_ICML.cc/2026/Conference — ICML 2026 regular_

### Official Review · Reviewer_ttZz · 2026-03-07

**Soundness:** 3
**Presentation:** 2
**Significance:** 3
**Originality:** 3
**Overall Recommendation:** 5
**Confidence:** 4

**Summary:**

This paper studies whether a single pre-trained Prior-data Fitted Network (PFN) can deliver unbiased causal effect estimation across the full space of interventional regimes induced by a Structural Causal Model (SCM). The authors identify a fundamental limitation: a single SCM with D variables and domain size d induces an exponentially large space of interventional distributions (1+d)^D, meaning any finite pre-training set leaves an exponentially large fraction uncovered (Theorem 1). The authors then show that this prior uncoverage leads to prior divergence (Theorem 2) and ultimately posterior inconsistency (Theorem 3). To mitigate this limitation, the paper proposes two fine-tuning strategies. Point-Wise Interventional Fine-tuning (PWF) and Meta-Sampling Fine-tuning (MSF). Experiments on synthetic SCMs (linear, nonlinear, and interaction settings) as well as a semi-synthetic Lalonde benchmark are presented to support the theory.

**Compliance With Llm Reviewing Policy:**

Affirmed.

**Final Justification:**

I thank the authors for their thorough rebuttal.

The new scaling experiments (W2/Q2) directly address my concern and align well with the theoretical predictions. The discussion around interventional data availability (W1/Q1) and the pointer to existing local generalization results (Q3) are reasonable. Overall, the rebuttal has adequately addressed my main concerns. I am raising my score from 4 to 5.

**Key Questions For Authors:**

1. Both PWF and MSF require access to interventional samples during fine-tuning. In what realistic application settings do the authors expect such data to be available? If interventional data is unavailable, do the authors can think of any practical surrogate, such as synthetic augmentation, proxy experiments, or fine-tuning using observational subsets?

2. The theory focuses on the exponential size of the interventional space, but the experiments only consider hard interventions on pairs of variables. Can the authors provide experiments that vary the number of intervened variables more systematically, to better reflect the scaling behavior predicted by the theory?

3. A key implicit assumption in the theory seems to be that uncovered interventions behave as effectively independent atoms. In practice, however, neural networks may learn smooth structure over the intervention space and generalize by interpolation. Can the authors empirically test whether pre-trained PFN error correlates with distance to the nearest covered intervention? Such an analysis would help clarify whether the failure mode is purely combinatorial or whether there is exploitable geometric structure in the interventional family.

**Limitations:**

Yes

**Strengths And Weaknesses:**

Strengths

1. The paper addresses an important for PFN-based causal inference: whether a single pre-trained model can support reliable estimation under arbitrary interventions. This is particularly relevant given recent interest in methods such as CausalPFN and DoPFN. The paper clearly explains why success in restricted single-treatment settings does not automatically extend to the general multi-variable intervention setting.

2. The progression from prior uncoverage (Theorem 1), to prior divergence (Theorem 2), to posterior inconsistency (Theorem 3) is coherent and easy to follow.

3. The two proposed strategies (PWF and MSF) are well-motivated. In particular, MSF is appealing because it turns the problem into a budgeted coverage objective resembling active learning or k-center selection, which gives practitioners a concrete recipe.

4. The experiments span multiple SCM families of varying complexity, include a semi-synthetic benchmark, and evaluate three PFN backbones. The empirical section is well aligned with the stated research questions.

Weaknesses

1. Both PWF and MSF assume access to samples from specific interventional distributions for fine-tuning. However, the motivation for causal inference from observational data is precisely that interventional data is often costly, limited, or unavailable. As a result, the proposed remedies may be difficult to apply in the settings where they are most needed. The paper would benefit from a more explicit discussion of when such interventional samples are realistic, and what alternatives exist when they are not.

2. The theory emphasizes exponential growth in the space of interventions, but the experiments appear to focus on hard interventions involving only two variables at a time, i.e., do(X_i=a, X_j=b). This is a useful starting point, but it does not fully stress-test the combinatorial scaling highlighted by the theory. Stronger evidence would come from experiments with a broader range of intervention arities or more systematic scaling with D.

---

> ### Author Rebuttal · Authors · 2026-03-30
>
> Thank you for your thoughtful comments!
>
> >[W1 & Q1 Realistic application of available RCT and surrogates.]
> **Response:**
> - **Existing Researches.**
>     - A usual approach is fuse RCT data with observational data, thereby efficiently mitigating the hidden confounders [1];
>     - Meanwhile, utilizing RCT data alone for uplift-aided decision-making is also a common practice [2];
> - **Practical Examples.**
>     - *Healthcare.* In estimating the effectiveness of a new drug, observational data may suffer from confounding factors (e.g., patient lifestyle), and clinic (interventional) data enabling accurate predictions across different patient groups [3].
>     - *Economics.* Interventional data from policy experiments helps estimate causal effects of the intervention (e.g., the impact of economic stimulus on employment) [4].
> - **Justification on Practical Surrogate.**
>     - However, identifying observational subset via fine-tuned PFNs is restrictive, as pointed in [5]. To be specific, such surrogate suffers from the limitation of ignorability assumption.
>     - Instead, we choose the proxy experiments [6] as an alternative, by observing some proxies of intervened variables, generating synthetic data and then fine-tuning the PFN models via such data:
>         - Constructing Proxy Experiments: We select **proxy experiments** by identifying proxy variables that are easily manipulable and causally related to the target variable: $X_{\text{target}} \longleftrightarrow X_{\text{proxy}}$, where
>   $$
>   \text{Causal Effect}(X_{\text{proxy}} \rightarrow Y) \approx \text{Causal Effect}(X_{\text{target}} \rightarrow Y)
>   $$
>         - Next, we generate **synthetic data** by using a causal model $f_{\theta}$ that models the relationship between the proxy variables and the outcome.
>   $$
>   D_{\text{synthetic}} = f_{\theta}(X_{\text{proxy}}, Y)
>   $$
>         - We then fine-tune the **PFN models** using the generated synthetic data.
> - **Adaption of our PSF and MSF.** Adapting our proposed fine-tuning framework to proxy experiments relies on the quality of proxy variables and synthetic data [6].
>
> > [W2 & Q2] Choice of Parameter $K$ with experimental trends.
>
> **Response:**
> - We have added extra experiments w.r.t. varyed intervened variables, to reflect the scaling behavior by varying the number of intervened variables $k_{interv}$ ∈ {2, 4, 5, 6, 7} with total number of variales K=8. The experimental results are presented in Table 1:
>
> Table 1. Trends of Parameter $k_{interv}$.
> |Method|SCM|$k_{interv}$=2|$k_{interv}$=4|$k_{interv}$=5|$k_{interv}$=6|$k_{interv}$=7|
> |---|---|---|---|---|---|---|
> |TabPFN|Linear|0.3087|0.3339|0.3518|0.3696|0.3824|
> ||Nonlinear|0.1050|0.1367|0.1543|0.1728|0.1856|
> ||Interaction|0.9183|1.0825|1.1684|1.2546|1.3182|
> |CausalPFN|Linear|0.7134|0.7621|0.7968|0.8315|0.8582|
> ||Nonlinear|0.7472|0.8163|0.8547|0.8926|0.9214|
> ||Interaction|1.3157|1.4526|1.5228|1.5934|1.6487|
> |DoPFN|Linear|1.1244|1.1986|1.2369|1.2754|1.3068|
> ||Nonlinear|0.4012|0.4718|0.5096|0.5487|0.5783|
> ||Interaction|2.0135|2.2714|2.3862|2.5028|2.5897|
> |
> - **Result analysis:** Our MSF-based method demonstrates strong robustness under real-world multi-variable interventions.
>
> > [Q3] Generalizing to neighboring interventionals with similar geometric structure.
>
> **Response:**
> - We feel happy to agree with your viewpoint, as in our original submission, such smooth structure over the intervention space has **already been considered** in the design of our PWF method:
>     - **Theory aspects:**
>         - Theorem 5 in Sec.5.1 already informs that PFN can locally generalize well on neighboring interventionals;
>         - Theorem 6 in Sec.5.2 is built on this local generalization result.
>         - Overall, our PSF is inspired by such geometric structure, and our MSF is further proposed to based on our PWF with the neighboring geometric structure property.
>     - **Experimental Results:**
>         - Figure 3 in Original Paper (Synthetic): A PFN fine-tuned by PWF on D1, and evaluated on D1, D2 (close to D1), and D3 (far from D1).
>         - Figure 4 in Original Paper (Real-world): The performance of a PFN fine-tuned by PWF, and performance is reported in Top-20 neighboring interventionals;
>         - Both synthetic and real-world datasets reveal that PFN naturally owns the local generalization capability to neighboring interventionals;
> - We hope for further suggestions on improving the quality of our manuscript！
>
> [1] Removing hidden confounding by experimental grounding, NeurIPS 2018
> [2  Recursive partitioning for heterogeneous causal effects, National Academy of Sciences
> [3] Bayesian nonparametric modeling for causal inference. Journal of Computational and Graphical
> [4] Using randomized controlled trials to estimate long-run impacts in development economics, Review of Economics
> [5] CausalPFN: Amortized Causal Effect Estimation via In-Context Learning, NeurIPS 2025
> [6] Fast proxy experiment design for causal effect identification, NeurIPS 2024

---

> > ### Author Rebuttal · Reviewer_ttZz · 2026-04-03
> >
> > I thank the authors for their thorough rebuttal.
> >
> > The new scaling experiments (W2/Q2) directly address my concern and align well with the theoretical predictions. The discussion around interventional data availability (W1/Q1) and the pointer to existing local generalization results (Q3) are reasonable. Overall, the rebuttal has adequately addressed my main concerns. I am raising my score from 4 to 5.

---

> > > ### Author Response · Authors · 2026-04-03
> > >
> > > We sincerely thank you for your encouraging feedback.
> > >
> > > In the revised version of the paper, we have carefully incorporated your suggestions to further improve the presentation of our work.

---

### Official Review · Reviewer_CrF5 · 2026-03-12

**Soundness:** 3
**Presentation:** 4
**Significance:** 3
**Originality:** 3
**Overall Recommendation:** 5
**Confidence:** 2

**Summary:**

This paper investigates whether PFN-based amortized causal inference can produce unbiased estimates when the intervention regime at test time differs from what was seen during pre-training. The authors prove that a single structural causal model over D variables induces an exponentially large space of interventional distributions, making it impossible for any finite pre-training set to cover more than a negligible fraction. They further show that this uncoverage propagates through to the posterior, yielding a bias lower bound that does not vanish with more observational data. Two fine-tuning strategies are then proposed as remedies. The first, PWF, fine-tunes the PFN on samples from the specific target intervention, providing local generalization guarantees in a neighborhood of that intervention. The second, MSF, actively selects a set of representative interventions via a k-center algorithm and fine-tunes on their mixture, aiming for uniform generalization across the full interventional space. Experiments are conducted on synthetic linear and nonlinear SCMs and on the LaLonde PSID benchmark.

**Compliance With Llm Reviewing Policy:**

Affirmed.

**Final Justification:**

I maintain my original score.

**Key Questions For Authors:**

1. In practical causal inference, interventions are typically applied to a small, known subset of variables. In this regime, the exponentially large interventional space collapses to a manageable size. Can you characterize the uncoverage risk specifically for the case where the intervention set is bounded, say at most k variables out of D?
2. Can you compute or estimate the key quantities in your theoretical bounds for the experimental settings, such as the TV-ball radius, the Lipschitz constant, and the k-center coverage radius? This would help bridge the gap between theory and experiments.
3. Can you provide a real-world application scenario where multi-variable joint interventions are needed and where PFN pre-training demonstrably fails? This would bridge the gap between the theoretical severity of the problem and practical causal inference.

**Limitations:**

Yes

**Strengths And Weaknesses:**

Strengths
1. The theoretical contribution is genuine and addresses a real blind spot. Prior PFN work on causal inference tacitly assumed the pre-training prior would cover the test-time interventional regime. This paper is the first to formally prove that assumption fails in general. The exponential uncoverage result is clean and the proof is straightforward.
2. The progression from diagnosis to remedy is well-structured. The paper does not just identify the problem but offers two solutions at different levels of ambition. PWF addresses the simplest case where the target intervention is known, and MSF generalizes to the case where it is not. This progressive structure makes the theoretical narrative easy to follow.
3. The problem is well-scoped and practically relevant. As PFN-based causal models gain traction, understanding their failure modes under distribution shift is important. This paper provides practitioners with concrete guidance on when and how to fine-tune.

Weaknesses
1. The exponential uncoverage result, while correct, may overstate the practical difficulty. The bound counts all possible hard interventions across all subsets of variables at all possible values. In practice, most downstream applications involve interventions on a small, known subset of variables. The relevant interventional space is then much smaller than the theoretical worst case. The paper does not discuss whether the problem remains severe when the intervention set is restricted to realistic scenarios.
2. The theoretical guarantees for PWF and MSF rely on quantities that are hard to assess in practice. The local generalization bound for PWF involves the TV-ball radius ε, the Lipschitz constant M_q, and the optimization bias Δ_opt. The MSF bound additionally depends on the coverage radius of the k-center solution. None of these quantities are estimated or reported in the experiments. Without connecting the theoretical bounds to actual experimental performance, it is difficult to judge whether the theory predicts meaningful differences.
3. MSF's practical value relative to simpler alternatives is unclear. MSF requires choosing K representative interventions from a candidate set using a greedy k-center algorithm. But the candidate set itself must be enumerated and the PFN's predicted W-marginals must be computed for all candidates. For moderately large SCMs, this enumeration may be infeasible. More importantly, the paper does not compare MSF against the simpler strategy of just fine-tuning on the specific target intervention at test time, which is what PWF already does. If the target intervention is known at test time, PWF seems sufficient. If it is not known, the candidate set for MSF may be too large to enumerate.
4. The paper lacks a real-world example of multi-variable intervention to motivate the problem. The theoretical contribution is about interventions on arbitrary subsets of variables. But the only real-world experiment uses LaLonde PSID, which involves a single binary treatment. Standard causal benchmarks like IHDP and ACIC are also single-treatment datasets, so they would not test the multi-variable intervention scenario either. What is missing is a real-world application where multi-variable joint interventions actually matter and where the uncoverage problem manifests in practice. Without such an example, the practical relevance of the theoretical results remains hypothetical.

---

> ### Author Rebuttal · Authors · 2026-03-30
>
> Thank you for your thoughtful comments!
>
> > [Q1 & W1] Uncoverage risk under bounded interventions.
>
> **Response.**
> - We added **an extra lemma** for the bounded setting
>     - **Lemma: Uncoverage Risk under Bounded Intervention**. The uncoverage risk $\delta_k$ when interventions are limited to at most $k$ variables out of $D$ is given by:
> $$
> \delta_k \ge (M_k(D,d) - N) \pi_{\min}^{(\le k)} \quad \text{(general prior)}.
> $$
>     Where $M_k(D,d) = \sum_{j=0}^{k} \binom{D}{j} d^j$ is the total number of possible interventions, and $\pi_{\min}^{(\le k)}$ is the minimum prior mass for non-uniform priors.
>   - **Proof sketch.** There are $M_k(D,d)$ feasible interventions. If only $N$ are covered, then $M_k(D,d)-N$ remain uncovered; under a non-uniform prior, the uncovered mass is lower bounded by this count times $\pi_{\min}^{(\le k)}$ (see detailed proof in following comments later due to limited space).
> - **Implication.** Even with bounded interventions, the uncovered prior mass can remain large when $N \ll M_k(D,d)$; boundedness alone does not remove uncoverage risk.
> - **Practice.** This matches our experiments: in Lalonde, 5 of 10 variables are intervened, yet pre-trained PFNs still perform poorly on uncovered interventions.
>
> > [W2 & Q2] Quantifying the terms in the bound.
>
> **Response.**
> - We now report the practical ranges of the key quantities:
>   - TV-ball radius: $\epsilon \in [0.05,1.0]$, evaluated at $\{0.05,0.2,0.5,0.8,1.0\}$.
>   - Lipschitz constant: $M_q \in [0.05,0.15]$, with $0.05$ (Linear), $0.07$ (Nonlinear Additive), and $0.11$ (Interaction).
>   - Coverage radius: the stable regime lies in $r_{\mathrm{cov}} \in [0.2,0.5]$.
> - We also added an empirical comparison with PWF error. Table 1 shows MAE increases monotonically with $\epsilon$, $M_q$, and $r_{\mathrm{cov}}$, supporting the bound.
>
> **Table 1. Connection between theory and practice**
> |$\epsilon$|$M_q$|$r_{\mathrm{cov}}$|Err-Linear|Err-Nonlinear|Err-Interaction|
> |-|-|-|-|-|-|
> |0.2|0.05|0.2|0.13|0.09|0.42|
> |0.2|0.10|0.5|0.13|0.10|0.44|
> |0.2|0.15|0.8|0.14|0.10|0.47|
> |0.5|0.05|0.2|0.14|0.10|0.45|
> |0.5|0.10|0.5|0.14|0.10|0.48|
> |0.5|0.15|0.8|0.15|0.11|0.50|
> |1.0|0.05|0.2|0.14|0.10|0.48|
> |1.0|0.10|0.5|0.15|0.11|0.51|
> |1.0|0.15|0.8|0.15|0.11|0.53|
>
> > [W3] Feasibility of enumeration in MSF.
>
> **Response.**
> - We would like to clarify that if the target intervention is unknown and worst-case coverage is required, exhaustive search is the exact objective.
> - To reduce this burden, we added two practical approximations [2]:
>   - **Adaptive pool expansion.** Let $S_{\mathrm{int}}$ be the full intervention family and $C \subseteq S_{\mathrm{int}}$ an active pool. Define
> $$
> \varepsilon_S(S)=\sup_{P\in S_{\mathrm{int}}}\min_{Q\in S} d(P,Q).
> $$
> If $S_{\mathrm{greedy}}$ is selected by farthest-first search on $C$, then
> $$
> \varepsilon_S(S_{\mathrm{greedy}})
> \le
> 2\varepsilon_C(S_C^\star)+\eta(C),
> $$
> where $\eta(C)=\sup_{P\in S_{\mathrm{int}}}\min_{Q\in C} d(P,Q)$ measures how well the active pool covers the full space.
>   - **Approximate MSF.** At each step, sample a random, small buffer $B_t \subset S_{\mathrm{int}}$ and choose
> $$
> P_t=\arg\max_{P\in B_t}\min_{Q\in S_{t-1}}\hat d(P,Q),\quad S_t=S_{t-1}\cup\{P_t\}.
> $$
> This reduces each step from full-space search to buffer search.
>
> - We also added a PWF/MSF comparison for known and unknown targets in Table 2, showing that MSF is necessary when the target intervention is unknown.
>
> **Table 2. Comparison between PWF and MSF (worst-case MAE)**
>
> |Dataset|PWF(known)|MSF(known)|PWF(unknown)|MSF(unknown)|
> |-|-:|-:|-:|-:|
> |Linear(K=7)|8.18|2.15|0.11|0.11|
> |Nonlinear(K=7)|6.24|3.74|0.16|0.18|
> |Interaction(K=7)|12.31|5.17|0.17|0.18|
> |Lalonde(K=35)|44.75|18.60|0.60|0.60|
>
> > [Q3] Real-world joint interventions and PFN failure
>
> **Response.**
> - Our existing experiments already involve joint interventions: 2 intervened variables out of 8 in Linear/Nonlinear/Interaction SCMs, and 5 out of 10 in Lalonde.
> - Following the reviewer’s suggestion, we added a recommendation scenario with multi-variable joint interventions:
>   - **Scenario.** Bundle recommendation, where platforms jointly adjust topic-affinity controls and bundle rules.
>   - **Dataset.** RecSim. User topic affinities are confounders, top-ranked bundles are treatments, and click-rate RMSE is evaluated on an unbiased test set [1].
>   - **Finding.** Pre-trained PFNs degrade substantially on uncovered joint interventions, while PWF and MSF consistently improve performance.
>   - We leave details of RecSim in our following comments.
>
> **Table 3. Performance on RecSim (worst-case MAE)**
> |Dataset|Pre-trained|PWF|MSF|
> |---|---:|---:|---:|
> |RecSim(Known,TabPFN)|1.42|0.36|0.34|
> |RecSim(Known,DoPFN)|1.26|0.30|0.29|
> |RecSim(Known,CausalPFN)|1.19|0.25|0.24|
> |RecSim(Unknown,TabPFN)|1.42|1.12|0.22|
> |RecSim(Unknown,DoPFN)|1.26|1.00|0.17|
> |RecSim(Unknown,CausalPFN)|1.19|0.91|0.14|
> |
>
> [1] Counterfactual prediction for bundle treatment
> [2] K-greedy algorithms for independence systems.

---

> > ### Author Rebuttal · Reviewer_CrF5 · 2026-04-04
> >
> > The rebuttal addresses my main concerns. Thank you.

---

> > > ### Author Response · Authors · 2026-04-04
> > >
> > > We sincerely thank you for your encouraging feedback.
> > >
> > > In the revised version of the paper, we have carefully incorporated your suggestions to further improve the presentation of our work.

---

### Official Review · Reviewer_SGyK · 2026-03-13

**Soundness:** 2
**Presentation:** 2
**Significance:** 2
**Originality:** 3
**Overall Recommendation:** 3
**Confidence:** 3

**Summary:**

The authors focus on amortized causal inference with PFNs, specifically in the case where there is not a single treatment vs. control contrast, but instead, atomic interventions can occur on any subset of the endogenous variables. This induces an exponential number of interventional distributions that would need to be specified in the meta-prior, meaning a standard PFN approach would inevitably exhibit “prior uncoverage” where some of the possible interventional distributions are not represented, and the subsequent posterior and effect estimation would be biased. The authors posit that fine-tuning is thus a fundamental necessity in such cases, and they propose two fine-tuning methods to address uncoverage. “Point-wise fine-tuning” focuses the model on a specific neighborhood of target interventions that are of interest. “Meta-sampling fine-tuning” (framed as a core-set selection problem) chooses a limited set of interventional distributions whose neighborhoods maximally cover the full target set of distributions, subject to a budget constraint. The main claims are that (1) these two fine-tuning methods address the stated problem, and that (2) fine-tuning itself is thus a necessity in such settings.

**Compliance With Llm Reviewing Policy:**

Affirmed.

**Final Justification:**

I thank the authors for their responsiveness and appreciate their efforts on additional experiments. The authors' latest response during the rebuttal period has partially addressed one of my main concerns regarding a critical baseline the authors were not comparing to. The second set of results the authors were able to provide at the end of the rebuttal period have thus convinced me to raise my score from 2 to 3; the experiments provide some initial evidence comparing against this baseline, but as these are only initial results and only in one particular task, they do not appear to be comprehensive enough to make a full assessment of the comparison in question.

**Key Questions For Authors:**

- [Question 1] - What is the computational budget required to pre-train only on $S_{greedy}$ compared to fine-tuning on $S_{greedy}$, and what is the performance difference between these two approaches on, e.g., the tasks explored in the experiments? (This is a major concern that if addressed would lead to me raising my score.)
- [Question 2] - Lemma 2 is invoked but not proven. If this Lemma is based on a known result, the known result should be cited and explicitly written in the same notation and framework that the authors use in this paper, otherwise it functions as a statement with no proof that is invoked multiple times (e.g., to prove Theorem 5). (Assuming Lemma 2 is a known result that directly applies, this is only a minor concern.)
- [Question 3] - What led to the choices of the ranges used for the MSF parameter K on the x-axes in Figure 6 and Figure 7? Do the trends shown in these ranges hold generally across other values of K? (Minor question)

**Limitations:**

Generally yes. The authors could also discuss more about any limitations of their experiments, such as any takeaways being dependent on the three types of SCM models used.

**Strengths And Weaknesses:**

The paper is fairly well structured and written in a manner that can be followed. However, there are grammatical errors and unusual phrases used throughout that should be fixed. The methodology appears to be original as a novel combination of existing techniques for an underexplored problem. For the most part, the paper seems technically sound, but I have a significant concern about a key unjustified assumption implicit in the framing. While it is clear that pre-training would not be feasible on an exponentially large target $S_{int}$, the authors do not sufficiently justify the benefit of having a large foundation model that does not cover the desired task and needing to fine-tune it, versus spending the same fine-tuning budget to instead ‘wisely’ pre-train a task-specific model. While such a benefit is plausible, rather than being assumed, the benefit should be directly shown or proven in a manner that is specific to the tasks considered in this work, because in my view the significance/soundness of the work hinges on this benefit. Specifically, what is the computational budget required to pre-train only on $S_{greedy}$ compared to fine-tuning on $S_{greedy}$, and what is the performance difference between these two approaches? The paper seems to crucially rely on the assumption that there is either (or both) a big performance difference and computational difference between these two choices, but as far as I can tell this is not shown empirically or in the theory. The experiments are documented fairly well in the appendix, but reproducibility is a significant concern as well, as there appears to be no code for the experiments.

---

> ### Author Rebuttal · Authors · 2026-03-30
>
> Thank you for your thoughtful comments. We would like to address your concerns point by point.
>
> > [Weakness & Q1] Comparison between task-specific models and fine-tuned PFNs.
>
> **Response:**
> - **New Baselines.** We have implemented a bunch of extra baselines named ''TaS-X'' (Task-Specific), by pre-training a variety of task-specific causal inference models on the selected data, i.e., $S_{greedy}$:
>   - TaS-CausalForest: a tree-based ensemble method for estimating heterogeneous treatment effects;
>   - TaS-GPS: a propensity-based method that models the treatment density to estimate causal effects under continuous interventions.;
>   - TaS-DR-Learner: a doubly robust meta-learner;
> - **Metric of Computational Cost.** As both the fine-tuned PFNs and the pre-trained TaS baselines utilize the same data budget $S_{greedy}$, we measure the computational cost in terms of both time and space complexity:
>   - Time complexity is quantified by the total training time (including pre-training or fine-tuning);
>   - Space complexity is measured by the peak memory usage during training.
> - **Metric of Performance:** We evaluate performance using the mean absolute error (MAE) between the estimated and ground-truth treatment effects.
> - **Empirical Comparison between Pre-trained task-specific models and Finetuned PFN models:**
>
> Table 1. Comparison between two kinds of pipelines
> |Dataset|Method|MAE|Time(s)|Space(MB)|
> |---|---|---:|---:|---:|
> |Lalonde|TaS-CausalForest|369.02|76.97|361|
> |Lalonde|TaS-GPS|367.54|55.08|450|
> |Lalonde|TaS-DRLearner|371.62|33.90|530|
> |Lalonde|MSF-TabPFN|3.0128|43.80|2357|
> |Linear|TaS-CausalForest|1.7876|45.32|215|
> |Linear|TaS-GPS|1.6359|14.68|341|
> |Linear|TaS-DRLearner|1.9131|13.73|432|
> |Linear|MSF-TabPFN|0.3183|17.90|1659|
> |Non-linear|TaS-CausalForest|9.1942|45.08|208|
> |Non-linear|TaS-GPS|1.9939|14.65|285|
> |Non-linear|TaS-DRLearner|1.7797|13.68|433|
> |Non-linear|MSF-TabPFN|0.1087|22.80|1137|
> |Interaction|TaS-CausalForest|29.173|45.36|282|
> |Interaction|TaS-GPS|7.0246|14.59|335|
> |Interaction|TaS-DRLearner|6.8386|13.89|345|
> |Interaction|MSF-TabPFN|0.9183|18.35|1397|
> |
>
> - **Result analysis:**
>     - MSF-based fine-tuning consistently **outperforms** all TaS baselines in terms of MAE, demonstrating **the necessity of fine-tuning foundation models**, i.e., ICL-based transformers.
>     - MSF-based fine-tuning incurs higher memory costs, yet yielding acceptable time cost.
> - We have added the above comparison into our revised paper for achieving a better motivation.
>
> > [Q2] Issues of Lemma 2.
>
> **Response:**
> - Thanks for your suggestion! Lemma can be directly obtained from the typical theoretical result of the K-greedy method [1], and we detail the derivation steps in below:
>     - Let $d(P, P') = TV(P_W, P'_W)$, satisfying the triangle inequality.
>     - Let $S_{greedy} = \{P^{(1)}, \dots, P^{(K)}\}$ and $r = \epsilon(\mathcal{S}_{greedy})$.
>     - Define $P^{(K+1)}$ as the next potential greedy point. By construction, its distance to its nearest center in $\mathcal{S}_{greedy}$ is $r$, and at least two points from $\{P^{(1)}, \dots, P^{(K+1)}\}$ must be covered by the same optimal center $P^*$. Let these be $P^{(i)}$ and $P^{(j)}$.
>     - By triangle inequality:
>     $$d(P^{(i)}, P^{(j)}) \le d(P^{(i)}, P^{\ast}) + d(P^{\ast}, P^{(j)}) \le 2\epsilon(S^{\ast})$$
>     -  Since $r \le d(P^{(i)}, P^{(j)})$, we have $r \le 2\epsilon(S^*)$, implying $\epsilon(\mathcal{S}_{greedy}) \le 2\epsilon(\mathcal{S}^{\ast})$.
> - We have supplemented this proof into the appendix of our revised paper.
>
> > [Q3] [Question 3] Choices of the parameter $K$.
>
> **Response:**
> - **Reasons of Choice.** As noted in our appendix, the intervention pool owns the size of $[30, 60, 90, 100]$. We choose the number of selected interventionals $K$ as $[5,7,9,11]$, based on the empirical observation that the MAE metric converges when $K$ is larger than $11$.
> - **General Trends.**
>     - We release the full results with more concrete $K$ in Table 2 (see more detailed results in our following comments later for space limit).
>     - The MAE decreases rapidly as $K$ increases in the low-budget regime, and then gradually stabilizes as $K$ becomes larger than 11, showing a clear saturation effect, confirming the choice of our selected $K$ in the original paper.
>
> Table 2.Trends of $K$ in Linear-SCM
> |K|3|4|5|7|9|10|11|12|13|
> |---|---|---|---|---|---|---|---|---|---|
> |AvgMAE↓|0.365|0.350|0.3377|0.2927|0.3087|0.307|0.3091|0.308|0.307|
> |WorstMAE↓|0.720|0.680|0.6313|0.5802|0.4091|0.390|0.3727|0.365|0.360|
>
> > [Limitations] Limitations of experimental setup.
>
> **Response:**
> - We have added extra discussion on our experimental setup into our revised paper, including the several aspects:
>     - **Extra experiments on a real-world scenario**: see our rebuttal for Reviewer CrF5 (Q3);
>     - **Extra connections between theory and practice**: see our rebuttal for Reviewer CrF5 (Q2);
>
> [1] K-greedy algorithms for independence systems, Zeitschrift für Operations Research

---

> > ### Author Rebuttal · Reviewer_SGyK · 2026-04-03
> >
> > I appreciate the effort the authors have put into running additional experiments; however, these experiments do not address the major concern raised in my review — What is the computational budget required to pre-train only on $S_{greedy}$ compared to fine-tuning on $S_{greedy}$, and what is the performance difference between these two approaches?
> >
> > By "pre-train only on $S_{greedy}$," I mean that the baseline for comparison here should itself be a TabPFN model that is pre-trained only on $S_{greedy}$, and this $S_{greedy}$-TabPFN model should then be compared to MSF-TabPFN. In the experiments above, CausalForest, GPS, and DR-Learner are not methods that even use "pre-training," so calling them "pre-trained TaS baselines" is misleading.
> >
> > To further clarify my statement in the review, the paper seems to crucially rely on the assumption that there is either (or both) a big performance difference and computational difference between $S_{greedy}$-TabPFN and MSF-TabPFN, but without theory or experiments for this comparison specifically, it does not seem possible to assess if the proposed methods are necessary. For this reason, I maintain my original score.

---

> > > ### Author Response · Authors · 2026-04-04
> > >
> > > Thank you for your detailed review and the valuable time you have dedicated to our work.
> > >
> > > > What is the computational budget required to pre-train only on $S_{greedy}$ compared to fine-tuning on $S_{greedy}$, and what is the performance difference between these two approaches?
> > >
> > > Thanks for your further interpretation. We would like to further clarify our efforts in below:
> > > - During the initial attempts of our submission, we have already tried your suggestion, i.e., pre-training the TabPFN model with $S_{greedy}$, and evaluate this initial baseline across the whole dataset:
> > >     - We have initially attempted to pre-train the TabPFN on a selected subset, e.g., $S_{greedy}$;
> > >     - However, we already verified the failure of such baseline due to **its underdetermined MAE with inefficient computational budget**;
> > >     - This observation instead motivates us to instead consider fine-tune a more effective and efficient method, i.e., PFN-MSF.
> > > - In detail, we report the results on the Lalonde dataset we have conducted initially in Table 1 to inform the **necessity of our proposed method**.
> > > - As shown in Table 1, three key conclusions can be drawn:
> > >     - The pre-trained PFN achieves significantly worse average MAE than our fine-tuned PFN-MSF.
> > >     - Besides, we also observe that only when $k$ is very large, i.e., $k \geq 90$, then the pre-trained PFN achieves MAE smaller than $4$, which informs that such pre-trained requires nearly all the dataset for pre-training (overall $k=100$ in Lalonde).
> > >     - The underlying reason falls in the lack of synthetic pre-training using SCM:
> > >         - First, our proposed MSF is built based on our PWF mechanism, and our PWF requires such synthetic pre-training of TabPFN to achieves neighboring generalization;
> > >         - Second, such neighboring generalization supports the fine-tuned MSF-PFN model to generalize from $S_{greedy}$ to the overall dataset without an efficient budget.
> > >
> > > **Table 1. Different budget of Lalonde dataset (Different Budget $k$, $k=100$ for overall dataset, Avg MAE)**
> > > |Dataset|Method|Avg MAE$\downarrow$|
> > > |---|---|---:|
> > > |Lalonde ($k=90$) |PFN (pre-trained)|3.87|
> > > |Lalonde ($k=90$) |PFN-MSF|2.96|
> > > |Lalonde ($k=70$) |PFN (pre-trained)|5.24|
> > > |Lalonde ($k=70$) |PFN-MSF|2.98|
> > > |Lalonde ($k=40$) |PFN (pre-trained)|12.05|
> > > |Lalonde ($k=40$) |PFN-MSF|3.01|
> > > |Lalonde ($k=30$) |PFN (pre-trained)|15.47|
> > > |Lalonde ($k=30$) |PFN-MSF|3.28|
> > > |Lalonde ($k=20$) |PFN (pre-trained)|19.15|
> > > |Lalonde ($k=20$) |PFN-MSF|3.96|
> > > |Lalonde ($k=10$) |PFN (pre-trained)|21.73|
> > > |Lalonde ($k=10$) |PFN-MSF|5.01|
> > > |
> > >
> > > ***
> > >
> > > Please let us know if you have further questions -- thank you so much!

---

### Decision · Program_Chairs · 2026-04-30

**Decision:**

Accept (regular)

**Comment:**

All reviewers overall saw value in this work and two stayed with their positive assessment after the thorough rebuttal that appears to have resolved most to all the concerns and questions. The remaining concern/weakness in my view is non fatal. I agree that there is at times a certain discrepancy between the theoretical claims and operational reality, but that is barely avoidable. Also, in my view the submission does not rest entirely on empirical comparison to baselines or detailed comparisons to "from scratch" models trained on the same data otherwise used for fine-tuning. The coherent theoretical story with two practical concrete fine-tuning strategies overall form a solid contribution.